# Stimulus encoding by specific inactivation of cortical neurons

Jesús Pérez-Ortega [1] ✉, Alejandro Akrouh [1] & Rafael Yuste [1]

Neuronal ensembles are groups of neurons with correlated activity associated with sensory, motor, and behavioral functions. To explore how ensembles encode information, we investigated responses of visual cortical neurons in awake mice using volumetric two-photon calcium imaging during visual stimulation. We identified neuronal ensembles employing an unsupervised model-free algorithm and, besides neurons activated by the visual stimulus (termed "onsemble"), we also find neurons that are specifically inactivated (termed "offsemble"). Offsemble neurons showed faster calcium decay during stimuli, suggesting selective inhibition. In response to visual stimuli, each ensemble (onsemble+offsemble) exhibited small trial-to-trial variability, high orientation selectivity, and superior predictive accuracy for visual stimulus orientation, surpassing the sum of individual neuron activity. Thus, the combined selective activation and inactivation of cortical neurons enhances visual encoding as an emergent and distributed neural code.

Neuronal ensembles have been defined as coactive neurons that repeatedly fire together[1], and are thought to be the functional unit of neural circuits underlying sensory perception, motor behavior, and memory[2–11]. In mouse visual cortex, neuronal ensembles encode visual stimuli[12,13] and are stable for many weeks[14]. However, the accuracy of neuronal ensemble representation of visual stimuli is still unclear. Hubel and Wiesel[15] showed that individual cortical neurons are tuned to specific features of visual stimuli, but trial-to-trial variability in neuronal responses during the same visual stimulus generate an unreliable estimation of visual features from single-neuron activity[16–19]. Using two-photon calcium imaging, recent studies have compared single-neuron to neuronal population encoding and found that populations better predict visual stimuli[20,21]. These studies were largely based on estimating prediction reliability from trained decoders using machine-learning algorithms, but did not explore the coding properties of neuronal ensembles. Here, we investigated how ensembles respond to drifting gratings using volumetric two-photon calcium imaging in primary visual cortex of awake mice. To analyze population neuronal activity, we used an algorithm[22] that identifies significant activity patterns, but without stimulus information. Such patterns were defined as neuronal ensembles. Interestingly, during each ensemble occurrence, not only was a group of neurons coactivated but also a distinct subset of neurons was also inactivated. We therefore redefined neuronal ensembles to include both coactive and inactivated neurons. We termed "onsemble" to the group of coactive neurons and "offsemble" to the group of inactivated neurons. Offsemble neurons were selectively inactivated during their preferred stimulus, as evidenced by a faster decay of their calcium signals. We quantified the orientation selectivity and tuning curves of neuronal ensembles as has been done for individual neurons[15–19,23,24]. We observed that ensembles exhibited lower trial-to-trial variability in response to visual stimuli, along with higher orientation selectivity and narrower bandwidth, as compared to tuned single neurons. These differences were not simply due to averaging the activity of coactive neurons (onsemble neurons). Even using an optimal activity threshold to predict the visual stimulus orientation, onsemble and offsemble neurons, as independent groups, were not as accurate as the ensemble. Our results indicate that cortical circuits can use a distributed neural code, where distinct neurons, selectively active and inactive − forming onsembles and offsembles, respectively − contribute to the encoding of visual information.

## Results

### Patterns of neuronal activation and inactivation during stimulus presentation

To investigate the neuronal dynamics underlying visual stimulus encoding, we performed in vivo volumetric two-photon calcium

[1]Neurotechnology Center, Dept. Biological Sciences, Columbia University, New York, NY 10027, USA. ✉e-mail: jesus.perez@columbia.edu

imaging in layer 2/3 of mouse primary visual cortex (V1). Animals were head-fixed and free to run on a wheel in front of a computer screen showing high-contrast drifting gratings along four different orientations in both directions (Fig. 1a). Mouse running speed and facial behaviors, recorded with an infrared camera, were tracked to measure wakefulness. Whisking behavior, reflecting mouse wakefulness or engagement, accounted for $61 \pm 3\%$ of the experiment duration. In contrast, when mice exhibited signs of disengagement or fatigue, they only whisked for $34 \pm 4\%$ of the time (Supplementary Fig. 1). We only analyzed mice that remained wakeful throughout the experiment ($n = 12$ mice). We recorded an average of $538 \pm 44$ active neurons per mouse (mean $\pm$ SEM; Fig. 1b, c). The location of V1 was confirmed with wide-field GCaMP imaging, and, on average, $371 \pm 31$ (69%) neurons were responsive to visual stimuli (mean $\pm$ SEM; Supplementary Fig. 2). To identify patterns of activity, we clustered all neuronal activity vectors − columns from raster plots − solely based on their similarity, without incorporating any information about visual stimuli, running speed, or whisking (Fig. 1d, e). A $z$-test was used to evaluate whether the similarity within a cluster of vectors was significantly larger than a random sample of vectors. If the similarity was significantly greater, the cluster was labeled as an ensemble, i.e., a pattern composed of neurons whose activity significantly changes, either because they

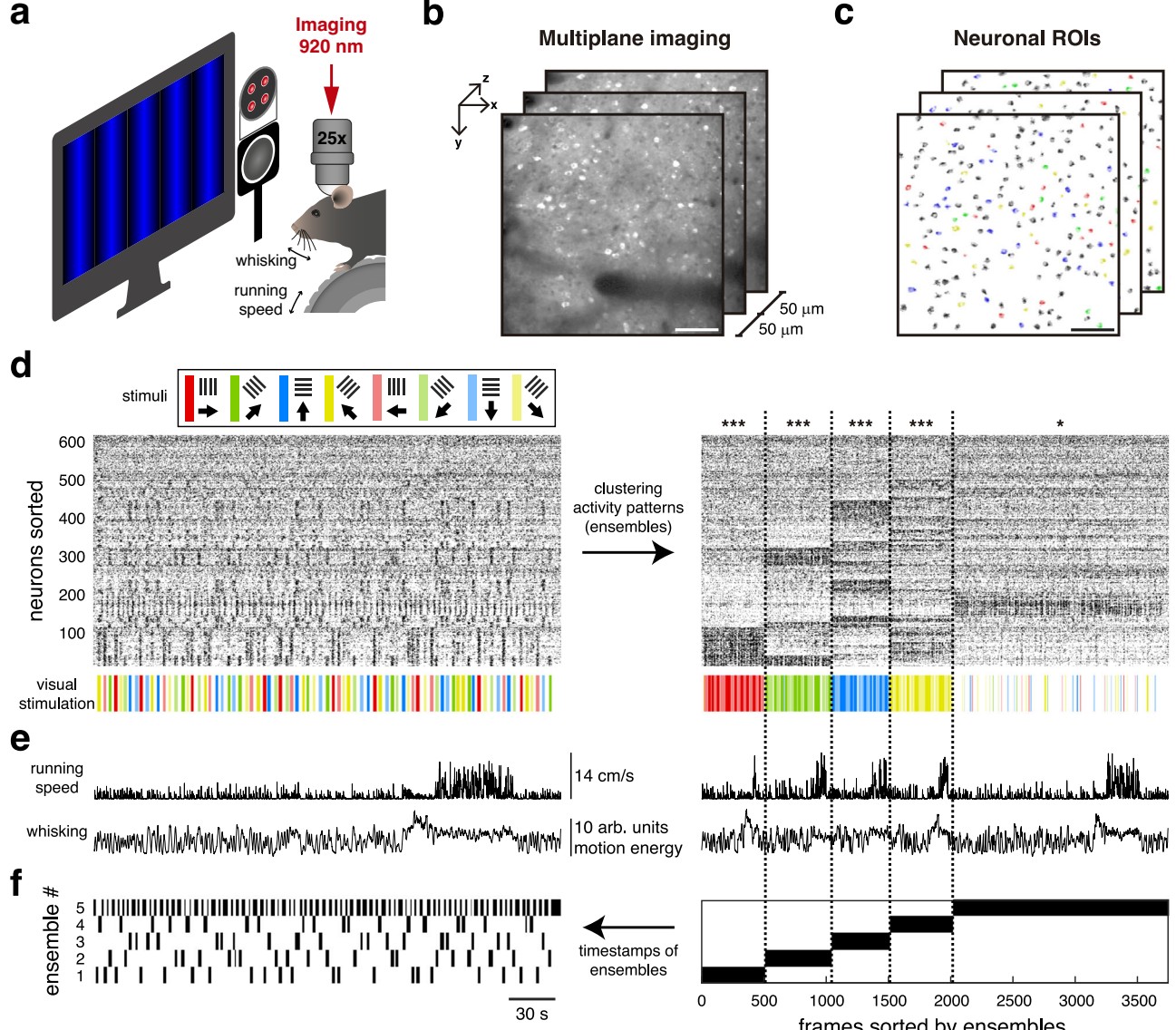

**Fig. 1 | Volumetric imaging of neuronal activity during visual stimulation and identification of activity patterns. a** Mice are head-fixed and GCaMP6s signals from pyramidal neurons are recorded through a cranial window via volumetric two-photon microscopy while visual stimuli are displayed on a monitor. Mouse running speed on a wheel and mouse whisking captured through an infrared camera are also recorded. Drawing of the experimental preparation adapted from Jesús Pérez-Ortega, Tzitzitlini Alejandre-García & Rafael Yuste (2021) Long-term stability of cortical ensembles eLife 10:e64449. https://doi.org/10.7554/eLife.64449. **b** Three sequentially recorded planes of layer 2/3 visual cortex at ~37 fps (i.e., a period of ~81 ms for 3 planes). Scale bar: 100 µm. **c** Neuronal regions of interest (ROIs) above 10 dB of peak signal-to-noise ratio (PSNR). Colored ROIs indicate neurons tuned to oriented drifting gratings (see Supplementary Fig. 1). Scale bar: 100 µm. **d** Raster plot of all neurons recorded simultaneously during a five-minute session of visual stimulation (left) accompanied by timestamps for random repetitions of drifting gratings along eight directions. Neurons are sorted by ensemble activity patterns, identified through clustering analysis applied to all frames (right). Stimulus presentation times are organized based on the clustering of activity patterns (bottom-right). Significance of each activity pattern's similarity within a cluster is tested using a one-sided z-test ($p < 0.5$). Significant activity patterns are identified as ensembles. Ensemble $p$ values from left to right $p = 3 \times 10^{-5}$, $p = 1 \times 10^{-4}$, $p = 5 \times 10^{-10}$, $p = 3 \times 10^{-5}$, $p = 0.04$. *$p < 0.05$, ***$p < 0.001$. **e** Mouse running speed and whisking motion energy (left), and they are sorted according to the activity pattern clustering detected in **d** on the right. **f** Ensembles timestamps (left) are extracted from the activity pattern clustering (right).

become selectively activated or inactivated by stimulus presentations. Finally, we identified the timestamps of ensemble occurrences and created a binary signal to indicate the active periods of each ensemble (Fig. 1f). This experimental setup and methodology allowed us to capture the activity of hundreds of neurons in V1 of awake mice during visual stimulation and extract ensembles along with their corresponding timestamps of occurrence.

In order to categorize the extent to which individual neurons participated in each ensemble, we defined an Ensemble Participation Index (EPI) as follows:

$$EPI = 2 \cdot \frac{FiringRate_{ensemble}}{FiringRate_{total}} - 1 \qquad (1)$$

where $FiringRate_{ensemble}$ is the fraction of time a neuron was active during all ensemble occurrences, and $FiringRate_{total}$ is the fraction of time a neuron was active during the entire experimental session. An EPI value of 1 means that the neuron was exclusively active during ensemble events. Conversely, an EPI value of −1 denotes that the neuron was active solely in the absence of ensemble events (i.e., the neuron was inactivated during ensemble events). An EPI of 0 implies that the neuron's activity remained similar both during ensemble activity and at other times. Using this index, we proceeded to assess whether a neuron exhibited significant activation or inactivation during ensemble events (See Methods, Fig. 2a, b). We defined the set of neurons displaying significant coactivation as onsemble neurons, the significantly inactive neurons as offsemble neurons, and those neurons that did not fall within the onsemble or offsemble group as nonparticipant neurons (Fig. 2b). The spatial distribution of onsemble and offsemble neurons exhibited heterogeneity (Fig. 2c). Neurons thus possess an EPI value for each ensemble, which means that a specific neuron might exhibit a positive EPI for one ensemble while simultaneously a negative EPI for another (Supplementary Fig. 3). Note that neurons can participate in multiple onsembles and offsembles, and nonparticipant neurons in one ensemble can be participants in another ensemble (Supplementary Fig. 3b). Next, we quantified the number of onsemble, offsemble, and nonparticipant neurons. Offsembles comprised a larger proportion of neurons compared to onsembles, with an average of $139 \pm 11$ offsemble neurons versus $108 \pm 9$ onsemble neurons for each ensemble ($20 \pm 1\%$ and $26 \pm 2\%$ of the total neurons, respectively). Additionally, $291 \pm 24$ neurons did not participate in the ensemble (counts represent means ± SEMs across 12 mice; Fig. 2d, e). The EPI is then used to distinguish whether neurons are activated or inactivated during ensemble events. Correspondingly, during ensemble events, significantly activated or inactivated neurons comprise the onsemble and offsemble, respectively. The remaining neuronal population was categorized as nonparticipant.

We subsequently analyzed the temporal evolution of neuronal activity during the onset of an ensemble (Fig. 2f) and quantified the average fraction of firing neurons within the onsembles, offsembles, and nonparticipant populations (Fig. 2g). The percentage of all active neurons prior to ensemble onset was $20 \pm 1\%$. At the onset of an ensemble event, we observed a gradual accumulation of active onsemble neurons, reaching a cumulative percentage of $95 \pm 1\%$ in two seconds. However, the peak fraction of simultaneously active onsemble neurons reached $58 \pm 2\%$ at $446 \pm 36$ ms after the ensemble onset. In contrast, the fraction of active offsemble neurons decreased to $8 \pm 1\%$ at $403 \pm 33$ ms after the ensemble onset, and the fraction of active nonparticipant neurons remained at $24 \pm 0.03\%$ (these values represent the means ± SEMs across 12 mice; Fig. 2e). Thus, during an ensemble event, its onsemble is activated while its offsemble is simultaneously inactivated, both reaching their maximum effect (activation and inactivation, respectively) between 400 and 450 ms after the ensemble onset.

## Offsemble neurons are specifically inactivated

We next examined the calcium signals of offsemble neurons to gain insight to the mechanisms of their suppression. We compared offsemble neuron decay time constants during spontaneous activity, interstimulus periods, and presentation of drifting gratings in its preferred orientation, which specifically inactivate offsemble neurons ($\theta_{pref}$; Fig. 3a, b). The preferred orientation of an offsemble is the same as that of the ensemble to which it belongs. Offsemble neurons showed no difference in decay time constants between spontaneous and interstimulus calcium transients ($5.8 \pm 0.3$ s and $5.9 \pm 0.2$ s, respectively). However, these decay time constants decreased to $2.5 \pm 0.1$ s during the offsemble's preferred stimulus orientation ($\theta_{pref}$; means ± SEMs across 7 mice; Fig. 3c). This suggests that individual offsemble neurons not only are silent during their preferred stimulus orientation but their calcium transients also decrease faster, indicating the possibility of an active inhibitory mechanism.

Furthermore, we evaluated the impact of locomotion on neuronal inactivation. We compared neuronal participation in ensembles between periods where the mice were still and when they were running (Fig. 4a–d). We observed no significant difference in the proportion of active neurons in ensembles during of quiescence or running ($24 \pm 1\%$ and $25 \pm 1\%$, respectively; means ± SEMs across 11 mice; Fig. 4e). While onsemble neurons did become more active during mouse locomotion, offsemble neuron inactivation appeared more pronounced, thus a consistent overall fraction of active neurons was maintained. To further quantify these differences, we measured the EPIs of onsemble and offsemble neurons during quiescence and locomotion. Onsembles participated equally between these two states ($0.41 \pm 0.02$ and $0.41 \pm 0.03$, respectively), whereas offsemble inactivation was significantly intensified when the mice were running (from $-0.46 \pm 0.03$ to $-0.52 \pm 0.04$; means ± SEMs across $n = 11$ mice; Fig. 4f). Although the peak fraction of simultaneously activated onsemble neurons can increase during running periods, their activity did not persist long enough to increase their EPI. In contrast, offsemble neuron inactivation persisted for longer periods (Fig. 4d), resulting in a significant change of EPI. We could depicted this subtle difference using EPI values. This observation suggests that during running periods, offsemble neurons exhibited accentuated inactivation rather than activation in onsemble neurons.

## Ensembles have higher orientation predicting precision than individual neurons

To compare the encoding capabilities of individual neurons and ensembles, we measured trial-to-trial variability, orientation selectivity, tuning curves, and prediction accuracy for visual stimulus orientation. First, binary signals were used for both neurons and ensembles, with each neuronal signal representing its spiking activity and each ensemble signal indicating its occurrences (Fig. 5a). We compared neurons and ensembles that were significantly tuned to a stimulus orientation. Ensemble responses showed minimal trial-to-trial variability to the preferred stimulus orientation ($\theta_{pref}$) occurring $91 \pm 3\%$ of the time at $490 \pm 51$ ms after the $\theta_{pref}$ onset. In contrast, individual neurons exhibited a response rate of $62 \pm 2\%$ at $379 \pm 39$ ms after the $\theta_{pref}$ onset (means ± SEMs across 12 mice; Fig. 5b). Additionally, while individual neuronal responses last $0.78 \pm 0.02$ s, ensemble responses persisted $1.61 \pm 0.04$ s (means ± SEMs across 12 mice; Fig. 5c). Thus, ensembles are more reliable and exhibit slower adaptation during a two-second period of visual stimulus compared to single neurons. Note that the precise timing of activation at the stimulus onset and the duration of activity may experience slight shifts or smoothing due to the filtering applied during calcium signal preprocessing for spike inference, potentially leading to minor inaccuracies.

Ensembles could have more information and a coding advantage simply because they are composed of many neurons. To investigate

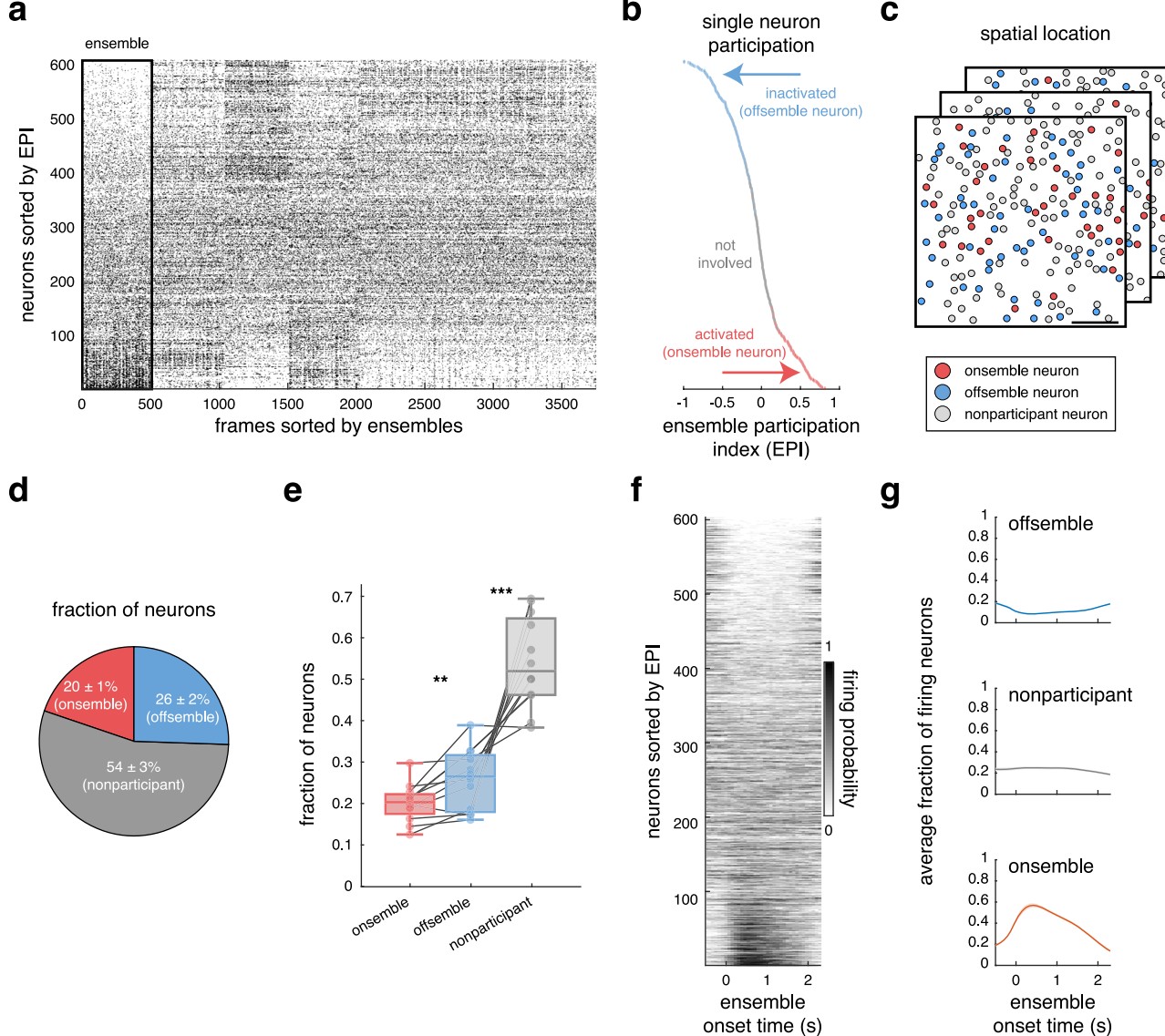

**Fig. 2 | Inactivation of neurons defines an offsemble. a** Representative raster plot with frames sorted according to ensemble identification as in Fig. 1b. A highlighted cluster represents the activity pattern of an ensemble, and neurons are sorted based on their specific participation in the ensemble activity pattern (**b**). This cluster contains all ensemble occurrences found in the recording. **b** Ensemble participation index (EPI) of each neuron for the highlighted ensemble in **a**. Colors indicate if neurons were significantly activated (red) or inactivated (blue) based on a two-sided *t* test (*p* < 0.05) while gray represents neurons without significant participation. **c** Spatial location of onsemble, offsemble, and nonparticipant neurons from raster in **a**. Scale bar: 100 μm. **d** Pie plot of the proportion of neurons belonging to an onsemble, offsemble, or nonparticipant population of ensembles found in *n* = 12 mice. Percentages are presented as mean ± SEM. **e** Average fraction of neurons that comprise an onsemble, offsemble, and nonparticipant population of all ensembles in a single mouse (*n* = 12 mice). The center of boxplots represents the median, the bounds of the boxes correspond to the first and third quartiles, and the whiskers extend to the minimum and maximum datapoint values. Two-sided Wilcoxon test: **\*\****p* = 0.005; \*\*\**p* = 9 × 10⁻⁴. **f** Average population activity of the ensemble onset activations highlighted in **a**. **g** Average fraction of active neurons in an onsemble, offsemble, and nonparticipant population during ensemble onset. Lines and their shades represent the mean ± SEM across *n* = 12 mice. Source data are provided as a Source Data file.

this, we measured the proportion of active neurons in onsembles (fraction of active onsemble neurons), offsembles (fraction of active offsemble neurons), ensembles (fraction of activated onsemble and inactivated offsemble neurons from onsemble and offsemble, respectively) and nonparticipant-tuned neurons (fraction of active neurons not part of the ensemble but tuned to visual stimuli; Fig. 5d). We compared the orientation selectivity and tuning curves between all of these groups. Ensembles, both as binary categories or as the fraction of participating ensemble neurons, exhibited a higher orientation selectivity of 0.98 ± 0.01 and 0.61 ± 0.05, respectively, than individual neurons (0.46 ± 0.01). Interestingly, the orientation selectivity of onsembles (0.23 ± 0.02) and offsembles

(0.20 ± 0.01) was lower than that of individual neurons. The nonparticipant-tuned neurons displayed the lowest selective (0.11 ± 0.01; means ± SEMs across 12 mice; Fig. 5e). Similarly, when fitting a Gaussian tuning curve to each group, ensembles showed a narrower bandwidth (binary: 8° ± 1°; fraction of participating ensemble neurons: 29° ± 3°) than individual neurons (37° ± 1°). Onsembles, offsembles, and nonparticipant-tuned group displayed broader tuning (52° ± 2°, 55° ± 1°, and 80° ± 2°, respectively; means ± SEMs across 12 mice; Fig. 5f, g).

To test if the binary spike detection biased the higher selectivity of ensembles, we repeated the analysis by measuring orientation selectivity and tuning width using spike inference data and obtained similar

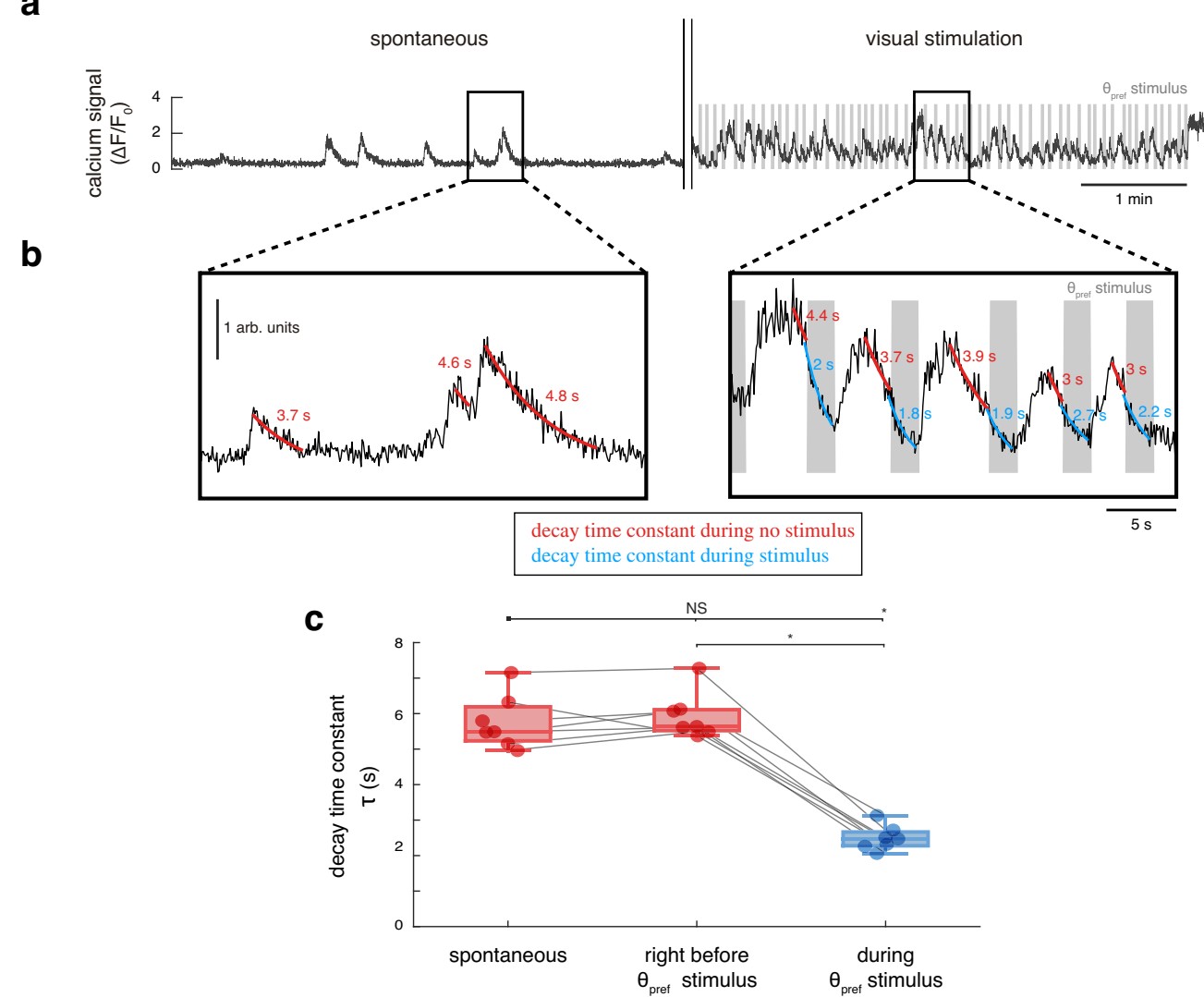

**Fig. 3 | Offsemble neurons have faster calcium decays during stimulus presentation. a** Example calcium trace of an offsemble neuron during five minutes of spontaneous activity (static blue screen, left), and five minutes of repeated presentation of its preferred inactivating visual stimulus ($\theta_{pref}$, right). **b** Magnification of selected calcium transients during spontaneous (left) and visual stimulation (right) sessions. The decay time constant ($\tau$) was obtained by fitting an exponential function for every spontaneous transient decay (red), every decay right before $\theta_{pref}$ onset (red), and every decay during $\theta_{pref}$ (blue). Exponential fits and decay time constants are overlaid on the calcium transients. **c** Average calcium signal decay time constants of offsemble neurons during spontaneous decay, decay right before $\theta_{pref}$ onset, and decay during $\theta_{pref}$. Red color indicates values during no visual stimulation, and blue color indicates values during visual stimulation. Two-sided Wilcoxon tests were performed (NS = not significant; *$p < 0.05$). Decay time constants remained similar with no visual stimulation ($p = 0.3$) but decreased during $\theta_{pref}$ onset ($p = 0.02$). Only mice with spontaneous activity recorded were included ($n = 7$ mice). The center of boxplots represents the median, the bounds of the boxes correspond to the first and third quartiles, and the whiskers extend to the minimum and maximum datapoint values. Source data are provided as a Source Data file.

results (Supplementary Fig. 4). Additionally, during the preferred stimulus orientation, the proportion of responding ensemble neurons was higher ($78 \pm 2\%$) than the proportion of tuned neurons ($49 \pm 2\%$) and the proportion of responding onsemble neurons ($47 \pm 2\%$; means ± SEMs across 12 mice; Supplementary Fig. 5). Therefore, the simultaneous activation and inactivation of neurons enhance ensemble orientation selectivity and tuning width.

## Ensembles have higher orientation predicting precision than onsembles

We further evaluated the accuracy of predicting visual stimulus orientation using confusion matrices, based on a set of four ensembles for each mouse, where each is associated to a specific orientation. In this analysis, we only compared data from mice containing the four necessary ensembles (11 out of 12 mice; Supplementary Fig. 6a).

Initially, to build the confusion matrix for individual neurons, we randomly selected four neurons, each tuned to a different orientation. For each neuron, we selected the optimal response threshold to obtain a binary signal representing the best prediction of its tuned orientation, and then we built a confusion matrix with the four tuned neurons. This process was repeated for the remaining groups of four tuned neurons, and the resulting confusion matrices were then averaged (neuron group, Fig. 6a). To build the confusion matrices for ensembles, onsembles, offsembles, and nonparticipant-tuned populations, with only four elements in each group, we selected the optimal response threshold to obtain a binary signal representing the best prediction and then we built the corresponding confusion matrices (Fig. 6a). Finally, we evaluated the accuracy of these confusion matrices for each mouse (Fig. 6b). Ensembles demonstrated superior accuracy ($1 \pm 0.002$) compared to individual sets of neurons ($0.61 \pm 0.02$).

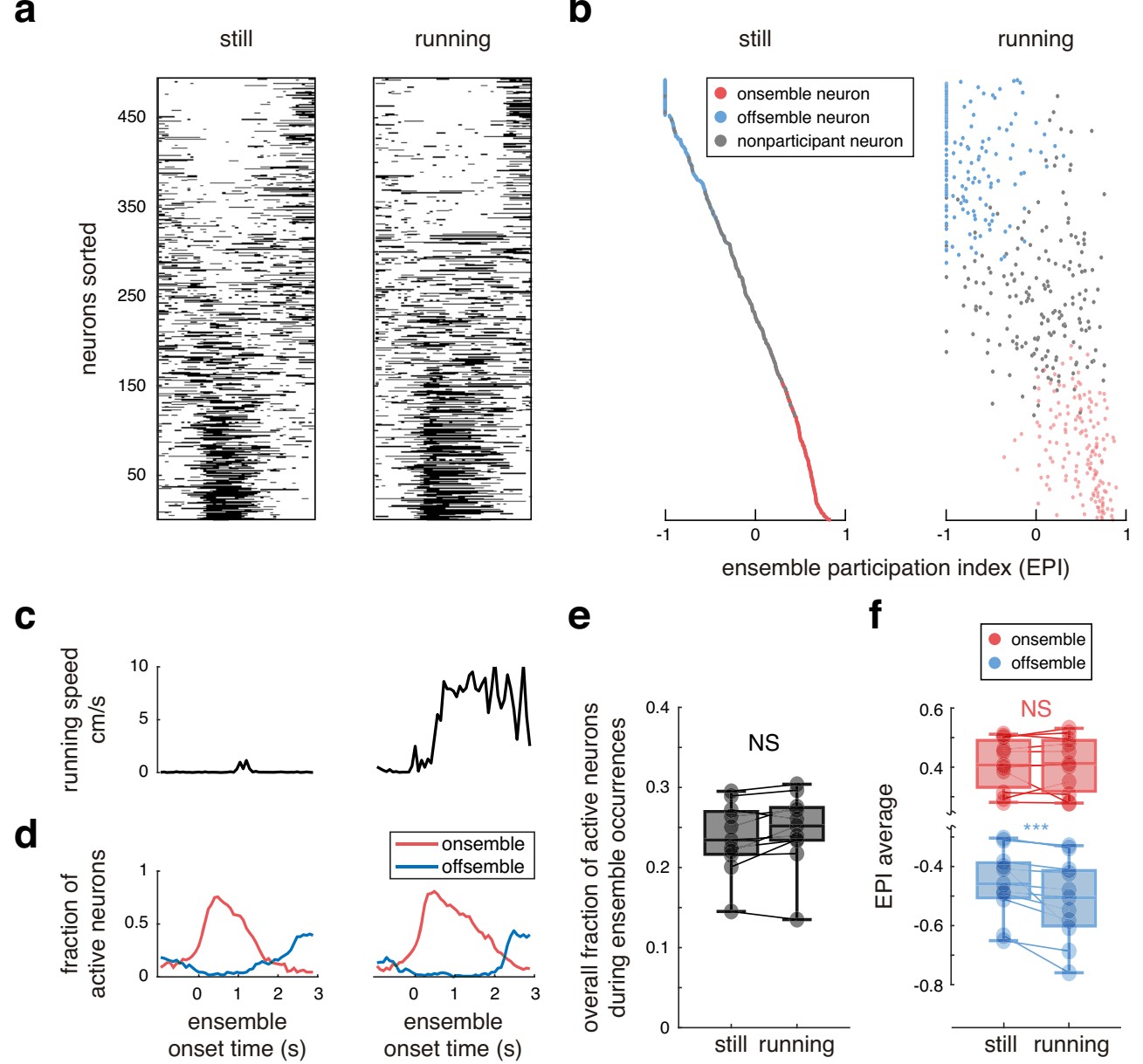

**Fig. 4 | Offsemble inactivation is enhanced during locomotion. a** Examples of the neuronal activity at the onset of ensemble activation when the mouse is still and running. Neurons are sorted based on their EPIs. **b** EPIs of onsemble (red), off-semble (blue), and nonparticipant (gray) neurons computed using the firing rate when the mouse was either still or running. **c** Locomotion traces for the example in **a**. Running periods where identified above 1 cm/s. **d** Corresponding fraction of simultaneously active onsemble (red) and offsemble (blue) neurons for the examples in **a**. **e** Fraction of active neurons during ensemble activity while the mice were still and running. Each datapoint represents the fraction of an individual mouse. Data were compiled from $n = 11$ mice that exhibited running speed >1 cm/s

during visual stimulation. Two-sided Wilcoxon test was performed ($p = 0.12$). **f** EPI averages of onsemble (red) and offsemble (blue) neurons for each mouse across periods of stillness and locomotion. Each datapoint represents the average for an individual mouse ($n = 11$ mice exhibiting running periods). Two-sided Wilcoxon tests were performed. Onsemble EPIs remained similar during stillness and loco-motion ($p = 0.6$), but Offsemble EPIs decreased during locomotion ($p = 9 \times 10^{-4}$). The center of boxplots represents the median, the bounds of the boxes correspond to the first and third quartiles, and the whiskers extend to the minimum and maximum datapoint values. NS not significant; ***$p < 0.001$. Source data are pro-vided as a Source Data file.

Onsembles and, surprisingly, offsembles also showed higher accuracy than individual sets of neurons ($0.89 \pm 0.03$ and $0.92 \pm 0.03$, respec-tively). In contrast, the nonparticipant-tuned populations displayed the lowest accuracy ($0.07 \pm 0.01$; means ± SEMs across 11 mice; Fig. 6b). This quantity may represent the fairest comparison between groups (neuron, ensemble, onsemble, offsemble, and nonparticipant-tuned population), given that all responses have been binarized to achieve the most accurate prediction possible. Consequently, ensem-bles exhibit greater accuracy than onsembles, indicating that they are not solely responsible for the precise predictions.

**Offsembles enhance orientation encoding of ensembles**

Finally, to test if offsemble neurons are responsible for the difference in precision between ensembles and onsembles, we reanalyzed the population activity removing offsemble neurons. Using the same algorithm to identify neuronal activity patterns, we found fewer ensembles encoding orientations, a significant reduction in their orientation selectivity and a broader tuning width (Supplementary Fig. 7). This effect did not happened when we removed the same number of nonparticipant neurons, even when the average of neurons removed was $60 \pm 3\%$ (mean ± SEM across 12 mice). Thus, the inclusion

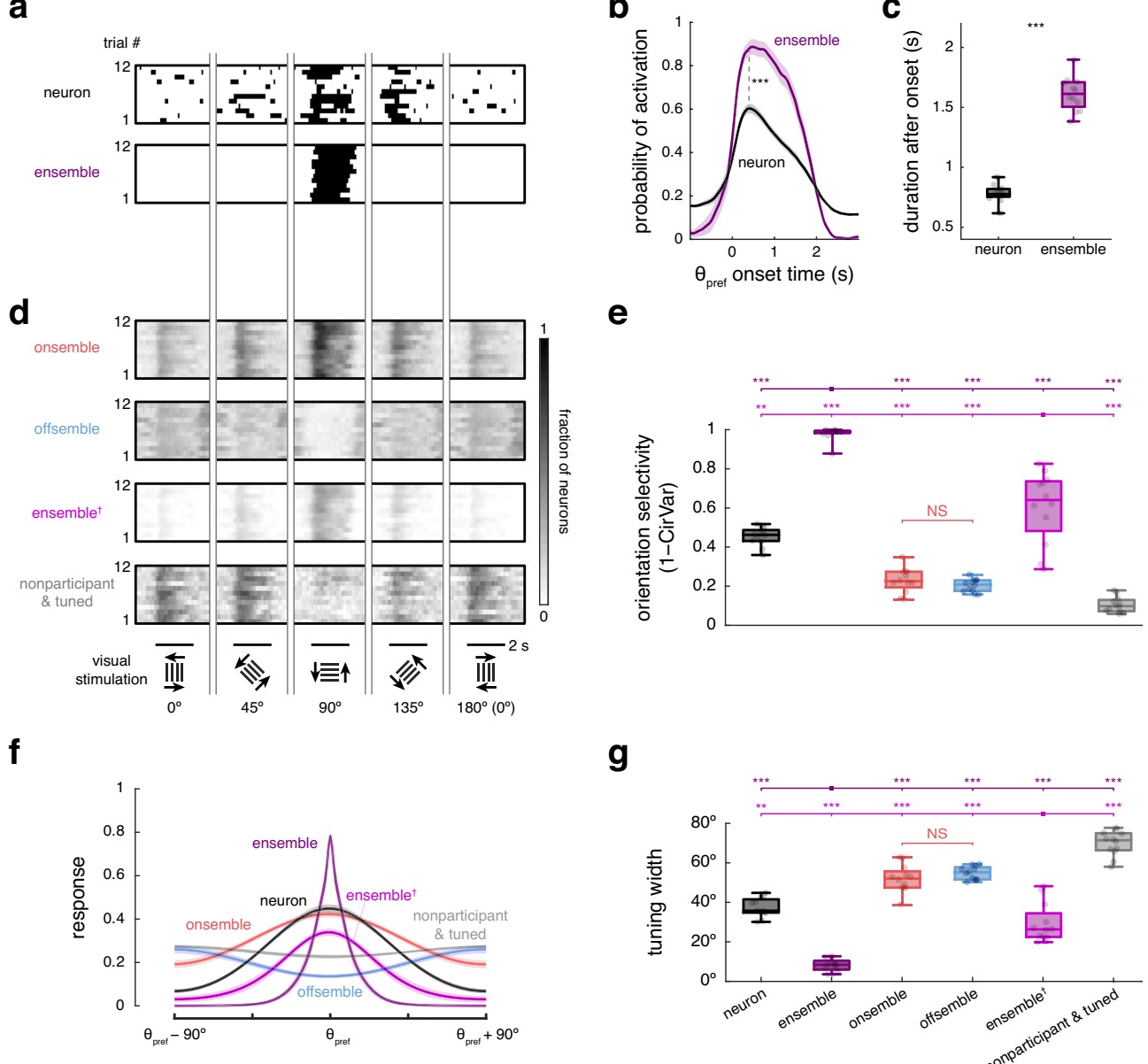

**Fig. 5 | Ensembles encode stimuli more accurately than individual neurons.**
**a** Example responses to four orientations for a neuron and an ensemble tuned to
the same preferred orientation (90°). For clarity, 0° trials are mirrored at 180°.
**b** Activation probability, representing the trial-to-trial variability of a neuron and an
ensemble activation at their preferred orientation ($\theta_{pref}$). Black dashed line indi-
cates maximum activation probability of individual neurons versus ensembles.
Two-sided Wilcoxon test, $p = 9 \times 10^{-4}$. **c** Duration of continuous evoked activity for
individual neurons and ensembles after the $\theta_{pref}$ onset. Two-sided Wilcoxon test,
$p = 5 \times 10^{-4}$. **d** Example responses across all trials for onsemble, offsemble,
ensemble[†], and the nonparticipant-tuned population, as in **a**. Onsemble, offsemble,
and nonparticipant-tuned population responses are presented as the fraction of
active neurons. Ensemble[†] responses are presented as the fraction of ensemble
neurons participating in the response (including activated onsemble neurons and

inactivated offsemble neurons). **e** Orientation selectivity. Ensembles showed higher
orientation selectivity than other groups (pairwise two-sided Wilcoxon test,
$p = 0.007$ or $p = 5 \times 10^{-4}$). **f** Orientation tuning curves fitted for individual neurons,
ensembles, onsembles, offsembles, ensembles[†], and the nonparticipant-tuned
population. Tuning curves are centered on $\theta_{pref}$. **g** Tuning widths, extracted from **f**.
Ensembles showed lower tuning width than the other groups (pairwise two-sided
Wilcoxon test, $p = 0.007$ or $p = 5 \times 10^{-4}$). Lines and shaded regions represent the
mean ± SEM across $n = 12$ mice. Each datapoint represents the average within each
mouse across $n = 12$ mice. The center of boxplots represents the median, the
bounds of the boxes correspond to the first and third quartiles, and the whiskers
extend to the minimum and maximum datapoint values. NS not significant,
**$p < 0.01$, and ***$p < 0.001$. Source data are provided as a Source Data file.

of offsembles neurons enhanced the precision of the ensemble. In
summary, ensembles are highly accurate in predicting drifting grating
orientations. While it might have been tempting to attribute this pre-
cision solely to the active neurons during ensemble activation
(onsembles), our findings showed that such high ensemble precision
was due to the emergent combination of both onsembles and
offsembles.

### Ensembles neurons have diverse tuning properties
To examine the tuning properties of the neurons that comprise
visually evoked ensembles, we categorized individual neuronal into
five response classes and measured their respective proportions within
onsembles, offsembles, and the nonparticipant population (Fig. 7a;
Supplementary Fig. 8). These five classes are neurons with: preference
for the same stimulus orientation (pref), preference for a different

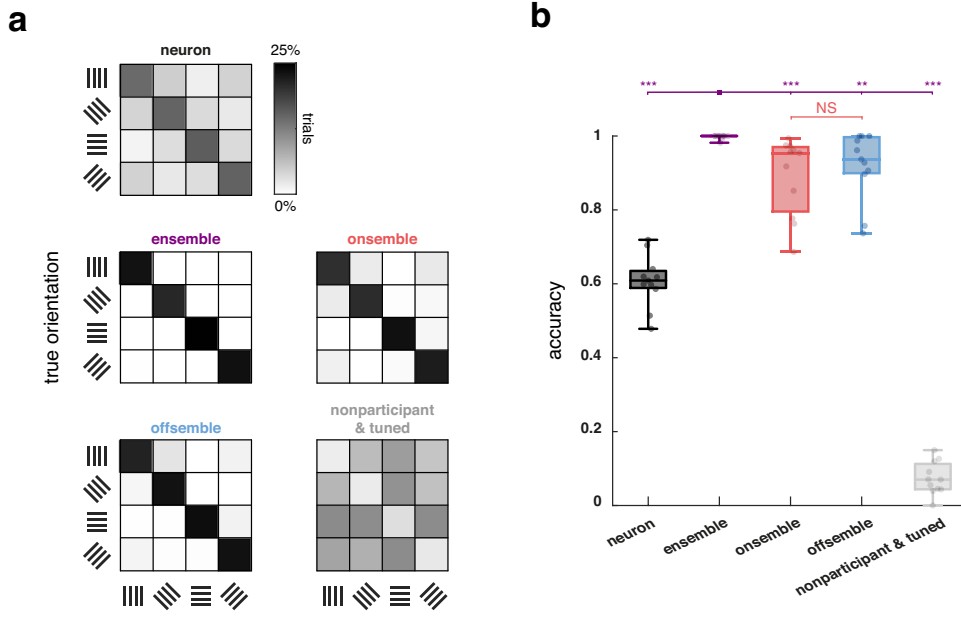

**Fig. 6 | Ensembles encode orientation better than onsembles or offsembles.** **a** Example of a mouse's confusion matrices used to predict the four orientations of visual stimuli for individual neurons, ensembles, onsembles, offsembles, and the nonparticipant-tuned population. **b** Accuracy in predicting the true orientation of visual stimuli using confusion matrices from individual mice. Each datapoint represents the accuracy of a single mouse using individual neurons, ensembles, onsembles, offsembles, and the nonparticipant-tuned population. Only mice showing four ensembles associated with the four orientations were included ($n = 11$ mice). Pairwise two-sided Wilcoxon tests were performed between groups. Ensemble accuracy was higher than any other group ($p = 0.008$ or $p = 9 \times 10^{-4}$). The center of boxplots represents the median, the bounds of the boxes correspond to the first and third quartiles, and the whiskers extend to the minimum and maximum datapoint values. NS not significant, $^{**}p > 0.01$, and $^{***}p < 0.001$). Source data are provided as a Source Data file.

stimulus orientation (nonpref), response to multiple stimuli (unspecific), preference for interstimulus periods (interstim), or no significant response to visual stimuli (untuned). In terms of onsemble composition, on average there were $44 \pm 3$ neurons tuned to the same stimulus orientation, $23 \pm 1$ neurons tuned to a different orientation, $26 \pm 4$ neurons with unspecific responses to multiple stimuli, and $14 \pm 2$ neurons that were not significantly tuned ($41 \pm 3\%$, $22 \pm 1\%$, $24 \pm 4\%$, and $13 \pm 2\%$ of onsemble neurons, respectively; means $\pm$ SEMs across 12 mice; onsemble on Fig. 7b). Thus, onsembles consist not only of coactive neurons tuned to the same orientation, but also of coactive neurons tuned to different or multiple orientations.

Offsembles were also composed of neurons with diverse tuning properties. On average, $63 \pm 6$ neurons were tuned to a different stimulus orientation, $51 \pm 8$ neurons were tuned to interstimulus periods, and $21 \pm 3$ neurons were not significantly tuned ($47 \pm 4\%$, $38 \pm 5\%$, and $15 \pm 2\%$ of offsemble neurons, respectively; means $\pm$ SEMs across 12 mice; offsemble on Fig. 7b). Thus, offsembles are comprised of suppressed neurons that are also tuned to a distinct orientation or to the interstimulus period. Conversely, nonparticipant population was composed of $128 \pm 11$ neurons that were not significantly tuned, $69 \pm 6$ neurons tuned to a different stimulus orientation, $45 \pm 3$ neurons tuned to interstimulus periods, $42 \pm 5$ neurons firing to multiple stimuli, and $6 \pm 1$ neurons tuned to the same stimulus orientation ($44 \pm 4\%$, $24 \pm 2\%$, $16 \pm 1\%$, $14 \pm 2\%$, and $2 \pm 0.4\%$ of nonparticipant neurons, respectively; means $\pm$ SEMs across 12 mice; nonparticipant population on Fig. 7b). Note that, per ensemble, a considerable portion of nonparticipant population is responsive to visual stimuli, so these neurons may be part of other ensembles.

Next, we analyzed the extent to which onsemble and offsemble neurons participate across multiple tuned ensembles. In a single ensemble, $58 \pm 5$ onsemble neurons ($54 \pm 4\%$ of the onsemble neurons) are shared with other onsembles, while $90 \pm 5$ offsemble neurons are shared with other offsembles ($65 \pm 3\%$ of the offsemble neurons; means $\pm$ SEMs across 12 mice, thin circles in Fig. 7b). The tuning properties of ensemble and offsemble neurons, along with the extensive sharing of neurons, illustrate how diverse neuronal responses are integrated in response to visual stimuli. In conclusion, our study reveals that cortical circuits respond to visual stimuli through a simultaneous activation and inactivation of multiple neurons with diverse individual tuning properties. Thus, ensemble coding represents an emergent property, activating onsemble neurons and inactivating offsemble neurons, that provides reliable and highly precise representation of visual information.

## Discussion

Here we report a neural coding strategy that combines the specific activation and suppression of neuronal responses. We investigated the population coding properties of cortical circuits in response to visual stimuli using volumetric calcium imaging of layer 2/3 from primary visual cortex in awake mice. Neuronal ensembles were previously described as a repetitive coactive group of neurons, but here we show that another selective group of neurons is simultaneously inactivated. Thus, we redefined neuronal ensembles as the combined activity pattern created by the activation and inactivation of two sets of neurons, termed onsembles and offsembles, respectively. Ensembles exhibit higher orientation selectivity and narrower bandwidth than individual neurons, along with a higher accuracy in predicting visual stimulus orientation. Furthermore, ensembles, by combining information about both activated and inactivated neurons, may integrate information more robustly and over longer periods for perception. While individual neurons are unreliable and adapt rapidly, which could lead to shifts in perception, the reliability and prolonged activity of ensembles may help stabilize representations of visual features[25–27]. Thus, neural responses to visual stimuli appear to be an emergent circuit property, where the activation and inactivation of groups of neurons enhance the precision and reliability of information processing.

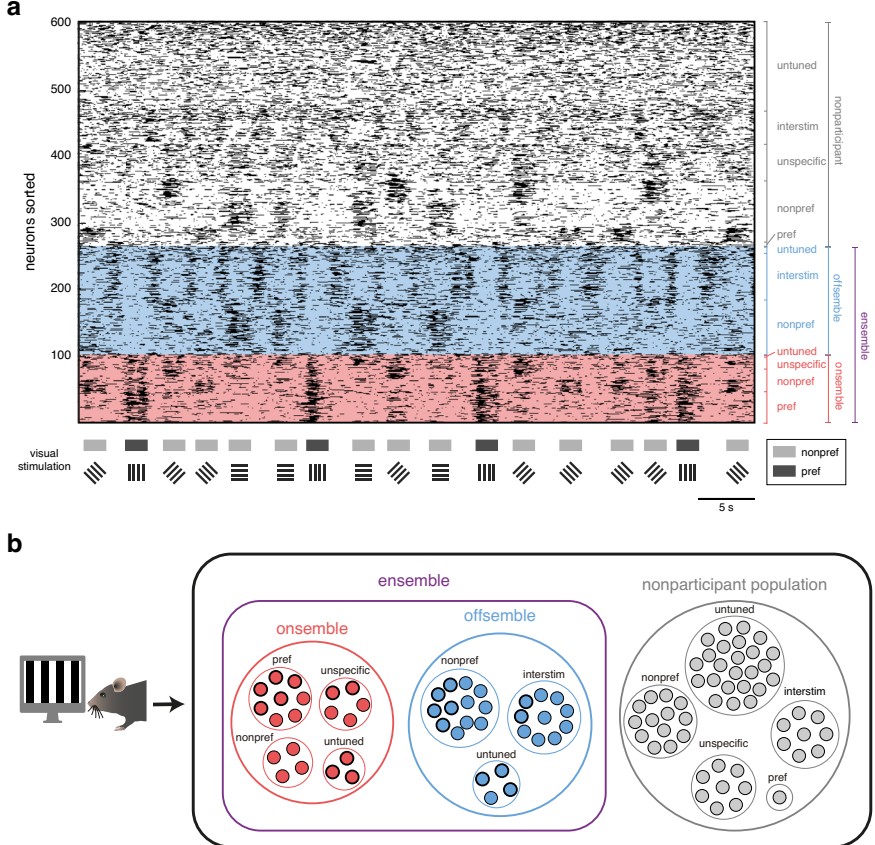

**Fig. 7 | Ensembles are composed of neurons with different tuning properties.**
**a** Example of a raster plot during visual stimuli highlighting an ensemble (onsemble-offsemble partnership) associated with a preferred orientation. Onsemble neuronal activity is represented in red shading, while offsemble neuronal activity is indicated in blue. Neurons are sorted and subdivided based on their individual tuning classification. Visual stimulation periods are indicated below raster plot, where pref represents the preferred ensemble orientation (in this example, 0°) and nonpref represents the other three nonpreferred orientations (in this example, 45°, 90°, and 135°). **b** Left, Drawing of a mouse visually stimulated adapted from Jesús Pérez-Ortega, Tzitzitlini Alejandre-García & Rafael Yuste (2021) Long-term stability of cortical ensembles eLife 10:e64449. https://doi.org/10.7554/eLife.64449. Right,

Venn diagram representing 100 pyramidal neurons in layer 2/3 of visual cortex during the onset of a visual stimulus presentation. Proportions were computed using data averaged from $n = 12$ mice. A group of neurons is activated (onsemble, in red), another group is inactivated (offsemble, in blue), and the third group remains uninvolved (nonparticipant population, in gray). Single-neuron analysis categorizes neurons as being tuned to the preferred orientation (pref), to a different orientation (nonpref), broadly tuned to multiple orientations (unspecific), associated with interstimulus periods (interstim), or untuned. Thick edges indicate neurons exclusive to the onsemble or offsemble of the preferred orientation while thin edges represent neurons shared within onsembles and within offsembles. Source data are provided as a Source Data file.

Identifying neuronal ensembles is a challenging problem and multiple methods have been developed to describe neural population activity, including thresholded correlations, template matching, principal component analysis, and graph theoretical approaches[4,12,14,24,28–32]. Here we introduce an unsupervised approach to detect neuronal ensembles, which not only identifies coactive neurons (onsembles), but also identifies inactivated neurons (offsembles). We developed a model-free algorithm[22] that, unlike previous methods[20,21,24], does not require training, additional external information, or optimization to maximize the prediction of stimuli. In contrast, our method (Figs. 1 and 2) identifies statistically significant repeated activity patterns in population vectors (ensembles), where each pattern includes an onsemble (active neurons) and an offsemble (inactivated neurons). As expected, due to biological variability between animals and experiments, the number of identified visual ensembles varied. In some instances, the number of cortical ensembles did not capture all visual orientations, while in others, there were also ensembles associated with stimulus directions (Supplementary Fig. 6c). Generally though, ensembles captured all 4 orientations presented (11 out of 12 experiments). Factors that may contribute to the variability in the number of visual ensembles include differences in neuronal count, signal-to-noise in the imaged field of view, and differences across

regions of primary visual cortex. Yet, even in datasets with less than 200 neurons, our method was able to detect ensembles encoding all presented orientations (Supplementary Fig. 6a).

We expected that ensembles tuned to a specific stimulus orientation would be comprised exclusively of neurons tuned to that orientation. However, many active (onsemble) neurons were either tuned to a different, or multiple, orientations and a small fraction was even untuned (Fig. 7). This diverse tuning properties of onsemble neurons explain why the orientation selectivity is lower than that of the neurons tuned to the ensemble orientation preference (Fig. 5e, g). In line with this finding, both nonspecific and untuned neurons have indeed been found to play a role in increasing a decoder's performance at predicting visual stimuli using machine-learning algorithms[33,34]. Moreover, the utility of heterogeneous tuning in neurons has been described before, such as in the motor cortex, where individual neurons make differentially weighted contributions to encode movement[35]. Thus, ensembles can be regarded as the emergent outcome of information processing of neurons with distinct roles, in agreement with the hypothesis that neuronal ensembles, rather than individual neurons, are the functional units of sensory encoding in cortex.

On the other hand, while specific subsets of neurons were activated to form onsembles, a complementary subset of neurons

(offsembles) were inactivated, further enhancing the cortical representation of visual inputs. Our results suggest that the inactivation is due to selective inhibition. Although commonly excluded from analysis, calcium decay signals can reveal synaptic inhibition[36–38], and, accordingly, we showed that calcium transients decay faster during the onset of their preferred inactivating stimulus (Fig. 3). Hyperpolarization during in vivo calcium imaging is known to reduce GCaMP fluorescence levels due to the correlation between intracellular calcium concentration and membrane polarization[36,39,40]. Previous studies have explored lateral inhibition and cross-orientation suppression during visual stimulation[41,42], but the role of inhibition in shaping single-cell activity has not been fully established[43–45]. Although inhibitory suppression may not be essential for single-cell tuning, it likely plays an important role for a population level coding, as has been proposed in models of orientation selectivity[46]. We hypothesize that multiple modes of inhibition shape offsemble neuron inactivation. For instance, a specific subset of offsemble neurons are tuned to the nonpreferred orientation, which may be due to mechanisms of cross-orientation suppression (Fig. 7a). Additionally, the subset of offsemble neurons tuned to interstimulus periods (interstim ensemble; ensemble 5 in Fig. 1f) may be the result of inhibition induced by grating stimuli, given that most neurons tuned to interstimulus periods were shared across all offsembles (interstim offsemble neurons in Supplementary Figs. 3 and 7). In summary, offsembles could be shaped by lateral inhibition, relief of excitatory drive, or neuronal intrinsic properties, all of which coordinate the inhibition that enhances visual coding in cortex[41,42,44,47].

Cortical inhibition is largely mediated by GABAergic neurotransmission and likely plays a central role in the specific inactivation of offsemble neurons. GABAergic interneurons shape the activity of pyramidal neurons and have been shown to enhance single-neuron orientation tuning. Conversely, optogenetic or pharmacological suppression of inhibition deteriorates tuning[48]. Local interneurons have been shown to be coherently active with excitatory ensembles[49] and suppression of parvalbumin-expressing (PV) interneurons reduces ensemble reliability during visual stimulation[50]. Furthermore, activation of vasoactive intestinal peptide-expressing (VIP) interneurons, which are highly correlated to locomotion, enhances weak responses to visual stimuli[51], perhaps also by augmenting offsemble suppression (as we show occurs during locomotion; Fig. 4). Thus, converging studies that support the role of inhibition in improving visual coding lead us to propose that GABAergic interneurons selectively target offsemble neurons. Disentangling the precise role of interneurons in shaping ensembles, specifically offsemble activity, presents exciting new avenues of inquiry.

Finally, recent studies have shown that animal locomotion alters neural activity in visual cortex. Locomotion can increase overall cortical activity[52], can be predicted from neuronal activity[10], and may enhance visual encoding[53,54]. In our experiments, locomotion had both effects: activation and inactivation of neurons. The total number of active neurons showed no significant global change when mice were running (Fig. 4e). However, running during the presentation of an ensemble's preferred orientation yielded an increase in offsemble neuron inactivation rather than an increase in the activation of ensemble neurons (Fig. 4f). Thus, locomotion may enhance stimuli encoding in cortex by further inactivating offsemble neurons.

In conclusion, our results show that mouse primary visual cortex exhibits precise encoding of visual stimuli by ensembles, which are sets of coordinated activity of both activated and inactivated neurons. We find evidence of potential selective inhibition of ensemble neurons, and demonstrate that their inactivation enhances encoding precision. Ensembles thus reflect an intricate partnership between the activation and inactivation of neurons, collectively enhancing the precision of information processing in visual cortex, as an emergent functional code.

## Methods

### Animals and surgery
All experimental procedures were conducted in accordance with the US National Institutes of Health and Columbia University Institutional Animal Care and Use Committee and were similar to our previous study[14]. Mice were housed in a controlled environment under a 12 h dark-light cycle at room temperature of ~23 °C and ~50% of humidity. Mice had ad libitum access to food and water. Experiments were performed in 12 adult transgenic mice (*Slc17a7*-IRES2-Cre, JAX stock # 023527) crossed with TIGRE2.0 Ai162 (TIT2L-GC6s-ICL-tTA2, JAX stock # 031562) maintained in C57BL/6 J congenic background. Mice were anesthetized with isoflurane (1.5–2%) while maintaining body temperature at 37 °C. Dexamethasone sodium phosphate (0.6 mg/kg) and Enrofloxacin (5 mg/kg) were administered subcutaneously and Carprofen (5 mg/kg) intraperitoneally. A titanium head-plate was attached to the skull, a 4 mm diameter craniotomy opened (center at 2.1 mm lateral and 3.4 mm posterior from bregma), and a round coverslip was implanted and sealed. Animals received postoperative Carprofen for two days. Mice were allowed to recover for five days with food and water ad libitum, and their health was checked daily.

### Visual stimulation
A custom-made MATLAB application[55] was used to display visual drifting gratings on an LCD monitor positioned 15 cm from the right eye at 45° to the long axis of the animal. In 10 out of 12 mice, only the blue channel of the monitor was used (red and green channels were disabled) to avoid light contamination in the photomultiplier[56] (PMT). No discernible difference was observed between blue and black/white drifting gratings across experiments when the imaging objective was shielded. Visual stimulation started with a blue screen (mean of grating luminescence) followed by full sinusoidal gratings (100% contrast, 0.13 cycles/deg, 2 cycles/s) drifting in 8 directions selected randomly (0°, 45°, 90°, 135°, 180°, 225°, 270°, and 315°) presented for 2 s, and a blue screen inter-gratings lasting between 1 and 5 s. Drifting gratings were presented at least 6 times for each direction during a 5-min session (at least 48 trials per experiment).

### Mouse facial recording
Mouse face was recorded using an infrared monochrome camera (DMK 21BU04.H, The Imaging Source) with a zoom lens (MVL7000, Navitar) and an infrared illuminator (AI4, Tendelux). Images were acquired at 15 frames per second and stored using IC Capture software (The Imaging Source). Whisking, blinking, and sniffing behaviors were measured using a custom-made MATLAB code[57].

### Volumetric two-photon calcium imaging
Imaging experiments were performed five days after head-plate implantation. Each mouse was head-fixed on a wheel under a two-photon microscope (a custom-modified Ultima IV, Bruker). Animals were acclimated to the head restraint for periods of 5–15 minutes for at least 2 days and exposed to visual stimulation sessions before recordings. The imaging setup was enclosed with a blackout screen to avoid light contamination into the PMT. An imaging laser (Ti:sapphire, $\lambda = 920$ nm, Chameleon Ultra II, Coherent) was used to excite GCaMP6s. The laser beam at the sample (30–60 mW) was controlled by a high-speed resonant galvanometer scanning an XY plane (256 × 256 pixels) at 17.7 ms (frame period) covering a field of view of 452 × 452 µm using a ×25 objective (NA 1.05, XLPlan N, Olympus). An electrically tunable lens (ETL) was used to change focus (z axis) during the recording. Imaging was performed using Prairie View Imaging (Bruker) in three planes (30–50 µm apart) recorded consecutively at a depth of 150 µm to 250 µm from the pia, pausing ~10 ms between planes for ETL focus to stabilize. Thus, we collected three frames, one per depth, every ~80 ms for 5 min. Imaging and ETL were controlled by Prairie View software.

## Neuronal activity extraction

Neural activity was analyzed entirely using custom-made MATLAB code[22] based on our previous work[14]. First, rigid translation motion correction from raw videos was performed, regions of interest (ROIs) were identified based on Suite2p[58], and a filtered calcium signal ($F_{filtered}$) from each ROI was extracted using the ROI raw signal ($F_{ROI}$) and its local neuropil raw signal ($F_{neuropil}$):

$$F_{filtered} = \frac{F_{ROI} - F_{neuropil}}{F_{neuropil}} \qquad (2)$$

Then, to evaluate these signals, peak signal-to-noise ratio (PSNR) was computed in decibels for each ROI as following:

$$PSNR = 20 \cdot \log_{10} \left( \frac{max\left(F_{ROI} - F_{neuropil}\right)}{std\left(F_{neuropil}\right)} \right) \qquad (3)$$

where max function is the maximum value and std function is the standard deviation. Subsequently, calcium signals above 10 dB of PSNR were selected and filtered sequentially by a median, minimum, and maximum moving filter of 500 ms window, which worked better for preserving timing. Then, we performed spike inference using the foopsi algorithm[59], followed by its binarization using a threshold proportional to the calcium signal's PSNR:

$$threshold = 0.04 - 0.002 \cdot PSNR \qquad (4)$$

The resulting binarized signal represents the presence or absence of spikes. All neuronal binarized signals were used to build a matrix of size $N \times F$, where $N$ is the number of active neurons, and $F$ the number of frames recorded. This matrix is visualized as a raster plot, where the ones in the matrix are the dots representing the spiking activity of the neurons (Fig. 1d).

## Ensemble activity detection

To identify ensembles, we used a custom MATLAB program[22], built upon our previous work[14]. First, we extracted the functional neuronal network from the raster matrix[14]. Subsequently, we filtered the raster by removing spikes from neurons that were nonfunctionally connected in each column vector. Following this preprocessing step, hierarchical clustering was applied to all column vectors using Jaccard similarity and Ward linkage (as illustrated in Fig. 1d). The optimal number of clusters was determined by identifying the maximum local contrast index[60]. Each cluster was considered an ensemble if the similarity between their vectors was statistically significant ($p < 0.05$), determined through a z-test. This significance was assessed by comparing the average similarity within the cluster vectors being tested to the mean and standard deviation average similarity among the same number of the cluster vectors selected randomly over 1000 iterations. Once significant ensembles were identified, their timestamps were located (ensemble activity, Fig. 1f). Note that, as our algorithm is an event-based method for ensemble identification, ensemble activity is a binary signal representing its occurrences, and the occurrences between ensembles are mutually exclusive.

## Identification of onsemble and offsemble neurons

We introduced the EPI to measure each neuron's involvement during ensemble activity, ranging from −1 to 1. A positive EPI indicates increased activity during ensemble activity, while a negative EPI indicates reduced activity. We determined EPI significance ($p < 0.05$) using a two-sample t-test, comparing the fraction of time that the neuron was active during ensemble activity vs ensemble inactivity. Neurons with significant positive EPIs were classified as onsemble neurons, and those with significant negative EPIs as offsemble neurons.

## Decay time constant

To assess potential inhibition in the calcium signals, we fitted a simple exponential function to single transient decays using the following equation:

$$decay(t) = a \cdot e^{-\frac{t}{\tau}} \qquad (5)$$

where $a$ is the amplitude of the signal, and $\tau$ is the decay time constant. Then, we compared the decay time constants during spontaneous and evoked activity, specifically, right before and right after the onset of a particular visual stimulus.

## Onsemble, offsemble, and nonparticipant-tuned population activity

For each ensemble, we computed the activity of its own onsemble and offsemble, and the subset of tuned neurons within the nonparticipant population (nonparticipant-tuned population). The activity was computed by averaging the binary signals of all individual neurons within each group (onsemble, offsemble, and nonparticipant-tuned population), i.e., the fraction of active neurons at every time point.

## Orientation selectivity and tuning width

The response for each neuron, ensemble, onsemble, offsemble, and nonparticipant-tuned population was computed by averaging their activity during each visual stimulation trial. Each response ranges continuously between 0 and 1, regardless of whether it is a neuron, ensemble, onsemble, offsemble, or nonparticipant-tuned population. The signals of neurons and ensembles are binary, individual neuronal activity represents spiking activity (0 = inactive; 1 = active), and ensemble activity represents ensemble occurrences (0 = no occurrence; 1 = occurrence). Thus, the average of neuronal and ensemble activity represents the fraction of time they were active during the two-second visual stimulation. The signals of onsembles, offsembles, and nonparticipant-tuned populations are nonbinary with values ranging between 0 and 1, each group represents the fraction of their active neurons at any sample (0 = no neurons of the group were active; 1 = 100% neurons of the group were active). Thus, the average of onsemble, offsemble, and nonparticipant-tuned population activity represents the average fraction of active neurons within the group during the two-second visual stimulation. Orientation circular variance (1−CirVar) was used to quantify orientation selectivity and the Hotelling's $t^2$-test was used to assess significance[23]. Significant orientation selectivity was used to assign tuning to neurons and ensembles, with each being assigned to one of the four presented stimulus orientations. Once identified tuned ensembles, we proceeded to extract the activity of their own onsemble, offsemble, and nonparticipant population and measure the orientation selectivity for each group. To compute the tuning width, responses to orientations were fit by nonlinear least squares optimization to the following Gaussian curve:

$$response(\theta) = a \cdot e^{-\frac{(\theta - \theta_{pref})^2}{2 \cdot \sigma^2}} \qquad (6)$$

where $a$ is the maximum response amplitude of the preferred orientation, $\theta_{pref}$ is the preferred orientation, and $\sigma$ is the tuning width. Coefficients were restricted to the following intervals: $a$ to the interval [0 1], $\theta_{pref}$ to the interval [−45° 180°], and $\sigma$ to the interval [1° 90°]. Additionally, we performed orientation selectivity and tuning width using the spike inference signals, i.e., before thresholding (Supplementary Fig. 4).

## Accuracy of prediction using multiclass confusion matrices

To evaluate the accuracy of predicting all four visual stimulus orientations, we used multiclass confusion matrices. We analyzed data from 11 out of 12 mice, each of which contained the four necessary ensembles,

each associated to a distinct orientation. Confusion matrices were constructed for individual neurons, ensembles, onsembles, offsembles, and the nonparticipant-tuned populations. Each confusion matrix involved four elements, corresponding to the four grating orientations, to predict all four orientations. In the case of neurons, we randomly assembled sets of four neurons, each neuron tuned to a different stimulus orientation. For each neuron, we averaged its response during each trial and built a receiver operator characteristic (ROC) curve by varying the threshold to its responses. This allowed us to evaluate its performance in predicting its tuned orientation. We selected the threshold that optimized performance on the ROC curve, maximizing the informedness (hit rate minus false alarm rate). Once we obtained the predicted orientation using the four neurons for every trial, we built the confusion matrix. This process was repeated for the remaining sets of neurons, and the confusion matrices were then averaged. Similar to the process of neurons, we built confusion matrices for ensembles, onsembles, offsembles, and the nonparticipant-tuned populations. In this process, we selected the optimal response threshold to maximize the informedness of the ROC curve for assigning predictions, and then generated the corresponding confusion matrices. Using the threshold to maximize informedness, we obtained binary signals that better predicted orientation tuning, regardless of the group analyzed (neuron, ensemble, onsemble, offsemble, and nonparticipant-tuned population). With these multiclass confusion matrices, we measured prediction accuracy by calculating it as the division of the total true positives (diagonal values) by the total number of trials.

### Removal of offsemble and nonparticipant neurons

To test whether offsemble neurons contribute to the high orientation selectivity and narrow bandwidth of ensembles, we reanalyzed the population activity by excluding offsemble neurons. To do so, we identified offsemble neurons for each orientation-encoding ensemble, and then subsequently eliminated them from the population activity. As control, we identified nonparticipant neurons for each orientation-encoding ensemble. Given that the total number of nonparticipant neurons exceeds that of offsemble neurons, we selected an equivalent number of offsemble neurons by choosing the least participant neurons. After removing offsemble (or nonparticipant) neurons from the population activity, we analyzed the remaining population activity to identify the same original number of neuronal activity patterns found (Supplementary Fig. 7a, b). It is worth noting that any offsemble neuron from one ensemble can be an onsemble or a nonparticipant neuron of a different ensemble. Thus, many offsemble neurons also served as onsemble or nonparticipant neurons in different ensembles. The same occurred for nonparticipant neurons, as any nonparticipant neuron from one ensemble can be an onsemble or offsemble neuron in a different ensemble.

### Single-cell tuning to multiple orientations, interstimulus periods, or untuned

In addition to orientation selectivity, neurons not tuned to any specific orientation were evaluated to see whether they were tuned to multiple orientations (unspecific) or tuned to interstimulus periods (interstim). Subsequently, samples were obtained measuring the average neuronal activity during two different periods: stimulation periods (independently of the orientation) and interstimulus periods. These samples were evaluated using a two-sample t-test, where significant neurons were labeled as unspecific if they were more active during stimulation periods and were labeled as interstim if they were more active during interstimulus periods. Nonsignificant neurons were label as untuned neurons. Using this test, we identified significantly active neurons during visual stimulation (unspecific), significantly active neurons during interstimulus periods (interstim), and neurons that did not exhibit significant changes in their activity between visual stimulation and interstimulus periods (untuned).

### Neuronal activity during locomotion

To measure the effect of running on neuronal activity, sessions were separated between still and running periods. Running periods were considered when the running speed was greater than 1 cm/s, otherwise they were considered still periods. Only 11 out of 12 mice had sufficient running periods for analysis (>5% frames). Then, the count of active neurons in the entire population was measured to compare between still and running periods, during visual stimuli or during no visual stimulation. The EPIs were also measured during still and running periods for all ensemble and offsemble neurons.

### Statistics and reproducibility

All statistical tests were performed using animal-wide summary statistics to assure independent observations and avoid nesting data[61–63]. Every single datapoint in boxplots represents the average within a mouse and is paired in different comparisons (Figs. 2e, 3c, 4e, f, 5c, e, g, and 6b). Traces and shaded regions represent the mean and SEM across mice (Figs. 2g and. 5b, f). We averaged single-neuron measures per stimulus orientation tuned, and then averaged again to get a single datapoint per mouse. For ensembles, we averaged the values of all tuned ensembles to get a single datapoint per mouse. Similarly, for onsembles, offsembles, and nonparticipant populations, we averaged the values to get a single datapoint per mouse. The representative images depicted in Fig. 1b from the recorded calcium imaging videos were reproduced multiple times. However, it is important to note that when replicating the experiment in the same mouse, the signal-to-noise ratio diminishes over weeks[14].

### Reporting summary

Further information on research design is available in the Nature Portfolio Reporting Summary linked to this article.

## Data availability

The source data underlying Figs. 2–7 and Supplementary Figs. 1, 4–7 are provided as a Source Data file. The experimental preprocessed dataset and results generated in this study have been deposited in the Dryad repository (https://doi.org/10.5061/dryad.w3r2280w7).

## Code availability

The source codes used in this research are available at the following Zenodo repositories: Xsembles2P[22] (https://doi.org/10.5281/zenodo.8423311), MouSee[55] (https://doi.org/10.5281/zenodo.7765050), and MoussionEnergy[57] (https://doi.org/10.5281/zenodo.8422691).

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

## Acknowledgements

This project was supported by The Revson Foundation (Grant No. 22-21) to J.P.-O.; NEI (F32EY029161) and NEI (K99EY033974) to A.A.; and NEI (RO1EY035248), NINDS (RM1NS132981), NIMH (RO1MH115900), NSF (2203119), and Vannevar Bush Faculty Award (ONR N000142012828) to R.Y. We thank lab members for helpful discussions. J.P.-O. dedicates this paper to the memory of Esteban Pérez-Botello.

## Author contributions

J.P.-O. conceptualized, designed and performed experiments, analyzed data, coded programs, and wrote the manuscript. A.A. conceptualized, designed and performed experiments, analyzed data, and edited the manuscript. R.Y. conceptualized and designed experiments, edited the manuscript, assembled and directed the team, and provided resources and equipment.

## Competing interests

The authors declare no competing interests.
