## [Peer Review File · Nature Communications]

Stimulus encoding by specific inactivation of cortical neuronsREVIEWER COMMENTS

Reviewer #1 (Remarks to the Author):

In this article, Perez-Ortega and colleagues used calcium imaging in awake mice to analyze the response of V1 neurons to the presentation of drifting gratings. Using a method developed by them in a previous study, they clustered the neuronal activity in patterns carried by several subpopulations of neurons called “ensembles”. They found that ensembles are better tuned than single neurons, and their activity encodes the visual stimulus more precisely. Moreover, ensembles are more active during the visual stimulus presentation, are active longer, have a sharper tuning curve, and better predict the visual stimulus orientation than individual neurons. They also found that some ensemble neurons, the “offensembles”, are suppressed during stimulus presentation. Those offensembles are as good as the ensemble neurons to predict the stimulus orientation. Yet, combining ensemble neurons and offensembles activity is even better for orientation prediction. Finally, they found that the tuning properties of neurons within an ensemble are diverse.

Altogether, the authors intend to demonstrate that ensembles (and offensembles) are functional computational units. However, they are not providing direct proof of the synergistic nature of the increase of information in ensembles. The improved encoding of the stimulus orientation found when comparing the tuning of ensemble and individual neurons could be due to the trivial addition of individual information contents. Indeed, it has been known for decades that more neurons contain more information leading to better predictions. Finally, the details of the methods are lacking in a way that not only prevents the reader from reproducing the analysis but also understanding the exact nature of the presented results.

Main criticisms

1. The choice of words very similar to define different concepts makes the paper very difficult to follow. Indeed, the word “ensemble” describes a pool of neurons but “offensembles” describes individual neurons whose active equivalent is “ensemble neuron”. The difference between “ensemble activity” and “ensemble neuron activity” also needs to be explained more clearly.

2. The methods are not described with enough detail so the reader can understand how ensembles are detected without extensive research in the team’s previous publications. Even after doing that, it seems that some changes were made in the algorithm since then, but it is hard to tell what. This absence of details prevents the readers from understanding what exactly an ensemble is and reduces their capability of assessing the novelty and physiological relevance of the results. For example, the criteria used to determine when an ensemble is active is unclear. From my understanding, an ensemble is active

when neurons that are part of it are active, in such a way that the population vector is similar enough to the identified pattern of the ensemble. But how does that translate in proportion of co-activated neurons? Seeing that 58% are activated on average during ensemble activation made me question the relevance of the ensemble activity.

3. The use of percentage across the whole manuscript dilutes/blurs the information about the actual number of neurons that contribute to the different proportions measured throughout.

4. The authors did not investigate if the better encoding provided by ensembles relies on actual synergy, or results from the mere summation of the ensemble neurons' individual probabilities in response to a given stimulus. Indeed, the sole result that several neurons carry more information than one is now already well established. It is therefore necessary to go further than this initial finding. From here, two important questions need to be addressed: [a] What is the trial-to-trial variability of ensembles and ensemble neurons? Could ensembles be detected if the trials were shuffled across ensemble neurons? How would such shuffling change the coding properties of the ensemble? [b] To prove the relevance of ensembles as a functional computational unit, it is necessary to determine that gathering a similar number of coding, but non-ensemble neurons, is not as efficient as true ensembles to improve the tuning and predictions of individual neurons.

5. Orientation is a circular parameter, so why did you fit the response of neurons and ensembles with a double Gaussian instead of circular wrapped gaussians or von mises? How did you solve the issue of neurons or ensembles with preferences at the edges (i.e., close to 0° or 360°)?

6. Some results should be further discussed. [a] What is the physiological interpretation of a longer response (Fig. 2f)? [b] How do you explain the steeper calcium decays (Fig 5)? To my knowledge, the time constant of the calcium trace is only due to the physical properties of the calcium reporter...

7. I did not understand how the ROC analysis was performed. How do you use a binary classifier system to classify 8 orientations? Did you do a ROC analysis for each orientation versus the rest and take the best?

8. In Figure 2e, how do you explain that the visually evoked response starts at $t=0$?

9. In Figure 6, it would be valuable to have an example of an ensemble showing all its neurons and their tuning curves.

10. Line 172: Comparing a fraction of active neurons and the percentage of time the ensembles are being active seems unfair given the different nature of the measures. Why do you expect the informedness to be comparable?

11. Line 194: How are the categories "interstim" and "untuned" mutually exclusive?

12. As a gaussian is fit on the responses of the neurons to the presentation of stimuli of 8 possible orientations, the preferred orientations can take any value between 0°-360°. What are the criteria for the decision that a neuron is tuned to the preferred orientation of an ensemble? This is particularly important to understand how you classify neurons within ensembles and offensembles.

Minor comments

1. Line 147: Please rephrase this sentence, the reader cannot understand what is compared to what.

2. Line 422 : zero, not cero

Reviewer #2 (Remarks to the Author):

Pérez-Ortega et al. examine orientation tuning ensembles (to first approximation, neurons with highly correlated activity) in L2/3 of V1 in GCaMP-expressing transgenic mice. Ensembles are detected using a custom algorithm. They then define an "ensemble participation index" which can detect not just ensemble neurons but also neurons that are effectively anti-correlated with the ensemble, or "offensembles". They show that decoding of orientation is better done when you combine information from both the ensemble and the offensemble. The idea is interesting and well presented and the experimental methodology is of high quality. The statistical analysis, however, is done improperly, basically because they pool across thousands of neurons in most cases which trivially makes P tiny. Given that n=7 mice (and 5 in some cases), it's likely but not certain that the existing data will suffice to do the statistics correctly. Given this situation, I do not feel confident in evaluating the other claims of the paper, since I simply do not know to what extent they are / are not valid, and will only comment on clarity/methodological issues in my review.

Major:

1. My largest concern is the statistical approach. To my understanding, in many analyses, the authors aggregate neurons across animals. Hence the $P < 10^{-78}$ values we see in some of the figures.

Aggregating neurons across animals and treating them as independent observations is improper because within animal, neurons co-vary, and strongly. Indeed, correlated and anti-correlated neurons are at the heart of this paper's claims, making aggregation particularly problematic. Aggregating is also bad because it is strongly biased by, potentially, a single animal with very large numbers of neurons or with a very large effect. I recommend the following papers for the authors, as they outline the problem nicely and succinctly: Aarts et al., 2015 Nat. Neurosci.; Aarts et al., 2014 BMC Neurosci; Galbraith et al. 2010 J Neurosci.

As explained in all the referenced papers, the appropriate approach is to either use animal-wide summary statistics (e.g., for Fig. 2C, compute the mean 1-CirVar for each group - neurons or ensembles - in each animal, then you will have $n=7$ measurements for tuned neurons and tuned ensembles) or a multi-level statistical analysis (this will slightly improve things beyond an $n=7$ - you may end up with power that you would get from 10 samples, for instance - but not thousands).

This problem is unfortunately present in almost every analysis - by figure panel, 2c, 2e, 2f, 2h, 2j, 3f, 4f, 5c, 6d, etc. .

The one place this was problem was not present was Fig 4e, where per-animal statistics were correctly used. Unfortunately, a Mann Whitney test was used with $n=5$ and no effect was found, from which the conclusion was drawn that locomotion had no effect. You will never have $p < 0.05$ with $n=5$ for Mann-Whitney. The test is inconclusive and the "conclusion" that things were not different between still and running cannot be made from the data.

2. This brings me to problem 2: in most of the statistical tests, n is not given, and it is not explained that all neurons/ensembles were aggregated across animals. Specify the n for every statistical test and the unit of n - and in the resubmission, this should in almost all cases be mice. If it is not mice, an appropriate multilevel test should be used and explained in detail in the methods.

3. Behavior is undercharacterized/underdescribed. It would be very useful to know what the animal is doing during most of the experiment. We get a hint of this in Fig.4 when locomotion is examined but as a reader I want to know the general statistics of behavior - how frequently is the mouse moving, are there long periods of standing still, does the animal groom, etc. Are animals being monitored for wakefulness? Or is running sufficiently frequent that the animal is clearly not asleep (if so, describe / show this). How frequently, in general, do animals move and run? Is the experimenter monitoring the animal with video?

Minor:

- Missing information: what is the threshold for ensemble membership. In the methods all that is said is that significant must be met with z-test on permuted data. Is this with a $p < 0.05$ threshold?

Response to Reviewers

Nature Communications manuscript NCOMMS-23-12936

We appreciate the time and effort invested in reviewing our manuscript. In response to the feedback, we have implemented significant revisions:

Abstract, Introduction, Results, and Methods Sections: We have revised the text to enhance the clarity and coherence of our study's core concepts.

Discussion: Additions have been made to the discussion to ensure alignment with the revised content.

Figures: We have revised our existing figures and added a new figure and two supplementary figures. We have also reordered figures for a better flow.

References: We have added more references to support our study.

REVIEWER COMMENTS

Reviewer #1 (Remarks to the Author):

In this article, Perez-Ortega and colleagues used calcium imaging in awake mice to analyze the response of V1 neurons to the presentation of drifting gratings. Using a method developed by them in a previous study, they clustered the neuronal activity in patterns carried by several subpopulations of neurons called “ensembles”. They found that ensembles are better tuned than single neurons, and their activity encodes the visual stimulus more precisely. Moreover, ensembles are more active during the visual stimulus presentation, are active longer, have a sharper tuning curve, and better predict the visual stimulus orientation than individual neurons. They also found that some ensemble neurons, the “offensembles”, are suppressed during stimulus presentation. Those offensembles are as good as the ensemble neurons to predict the stimulus orientation. Yet, combining ensemble neurons and offensembles activity is even better for orientation prediction. Finally, they found that the tuning properties of neurons within an ensemble are diverse.

Altogether, the authors intend to demonstrate that ensembles (and offensembles) are functional computational units. However, they are not providing direct proof of the synergistic nature of the increase of information in ensembles. The improved encoding of the stimulus orientation found when comparing the tuning of ensemble and individual neurons could be due to the trivial addition of individual information contents. Indeed, it has been known for decades that more neurons contain more information leading to better predictions. Finally, the details of the methods are lacking in a way that not only prevents the reader from reproducing the analysis but also understanding the exact nature of the presented results.

We appreciate the thoughtful feedback provided, and have addressed each concern to enhance the clarity, detail, and quality of our manuscript.

Main criticisms

1. The choice of words very similar to define different concepts makes the paper very difficult to follow. Indeed, the word “ensemble” describes a pool of neurons but “offsemble” describes individual neurons whose active equivalent is “ensemble neuron”. The difference between “ensemble activity” and “ensemble neuron activity” also needs to be explained more clearly.

Response: We greatly appreciate this critique and we agree that the terminology was confusing. We have implemented changes that fully address this and enhance the clarity of our work.

To clarify the confusion, we have redefined the term ensemble to the pattern of activity that includes simultaneously active and silenced neurons. We now introduce the term “onsemble” to replace “ensemble neuron activity”. Specifically, “onsemble” refers to the group of active neurons during an ensemble occurrence while “offsemble” refers to the group of silenced neurons during an ensemble occurrence. These adjustments are reflected throughout the manuscript.

We have added the following text in the abstract:

“We detected neuronal ensembles employing an unsupervised model-free algorithm. [...] During an ensemble occurrence, besides neurons that were significantly activated by the stimulus, we also found neurons whose activity was significantly decreased in response. To distinguish between these two groups of neurons, we introduced the term “onsemble” for the significantly active neurons and “offsemble” for the silenced ones.”

We have added the following text in the introduction:

“Interestingly, during each ensemble occurrence, not only a group of neurons was coactivated but also a distinct subset of neurons was inhibited. We therefore redefined neuronal ensembles to include both coactive and silenced neurons. We termed the group of coactive neurons an “onsemble” and the group of silenced neurons an “offsemble”.”

We have added the following text in the results:

“Ensembles exhibit higher precision predicting drifting grating orientation than the integration of coding neurons

[...] we extended our analysis to include comparisons involving the activity of different neuronal groups (Figure 3d): the onsemble (fraction of active onsemble

neurons), offensemble (fraction of active offensemble neurons), and nonparticipant-tuned group (fraction of active neurons not part of the onensemble or offensemble but tuned to visual stimuli).”

2. The methods are not described with enough detail so the reader can understand how ensembles are detected without extensive research in the team’s previous publications. Even after doing that, it seems that some changes were made in the algorithm since then, but it is hard to tell what. This absence of details prevents the readers from understanding what exactly an ensemble is and reduces their capability of assessing the novelty and physiological relevance of the results. For example, the criteria used to determine when an ensemble is active is unclear. From my understanding, an ensemble is active when neurons that are part of it are active, in such a way that the population vector is similar enough to the identified pattern of the ensemble. But how does that translate in proportion of co-activated neurons? Seeing that 58% are activated on average during ensemble activation made me question the relevance of the ensemble activity.

Response: We have enhanced the clarity of our methodology providing a more detailed description in Methods, Figure 1, and Figure 2.

To maintain consistency with our terminology explained in Comment 1, we have renamed the method section “**Identification of onensemble and offensemble neurons**” to clarify how we defined these components.

Furthermore, we introduced a new section in the methods “**Onensemble, offensemble, and nonparticipant-tuned population activity**”, which explains how we extract and analyze the activity of each neuronal group.

Again, we now refer to the significant clusters of activity patterns as “ensembles”, which each one includes its own onensemble and offensemble. We also have added that, while the percentage of simultaneous active onensemble neurons on a single frame is approximately 58 %, the cumulative percentage of active neurons can reach up to around 95 % of the total onensemble neurons over time.

In our results section, we have included the following clarification:

“Patterns of neuronal activation and silencing during stimulus presentation

[...] At the onset of an ensemble event, we observed a gradual accumulation of active onensemble neurons, reaching a cumulative percentage of $95 \pm 1\%$ in two seconds. However, the peak fraction of simultaneously active onensemble neurons reached $58 \pm 2\%$ at 446 ± 36 ms after the ensemble onset.”

3. The use of percentage across the whole manuscript dilutes/blurs the information about the actual number of neurons that contribute to the different proportions measured throughout.

Response: We appreciate this observation. To ensure clarity and transparency, we have included the actual number of neurons and their proportions throughout the entire manuscript.

4. The authors did not investigate if the better encoding provided by ensembles relies on actual synergy, or results from the mere summation of the ensemble neurons' individual probabilities in response to a given stimulus. Indeed, the sole result that several neurons carry more information than one is now already well established. It is therefore necessary to go further than this initial finding. From here, two important questions need to be addressed: [a] What is the trial-to-trial variability of ensembles and ensemble neurons? Could ensembles be detected if the trials were shuffled across ensemble neurons? How would such shuffling change the coding properties of the ensemble? [b] To prove the relevance of ensembles as a functional computational unit, it is necessary to determine that gathering a similar number of coding, but non-ensemble neurons, is not as efficient as true ensembles to improve the tuning and predictions of individual neurons.

Response: Thank you for bringing these important points to our attention.

[a] We have specified the trial-to-trial variability in neurons and ensembles in the results and it is displayed in Figure 3b. We have added the following text:

“Ensembles exhibit higher precision predicting drifting grating orientation than the integration of coding neurons

[...] Ensemble responses showed minimal trial-to-trial variability to the preferred stimulus orientation (θ_{pref}) occurring 91 ± 3 % of the time at 490 ± 51 ms after the θ_{pref} onset. In contrast, individual neurons exhibited a response rate of 62 ± 2 % at 379 ± 39 ms after the θ_{pref} onset (means \pm SEMs across 12 mice; Figure 5b).”

We have performed the shuffling suggested and ensembles were detected similarly, without any major change in coding properties. It seems that this shuffling just equalizes the level of participation of each ensemble neuron on each trial.

We also provide below a non-manuscript figure with an example of the same experiment in Figure 1 with the activity shuffled within ensemble neurons tuned to orientations and the result of their hierarchical clustering for ensemble detection.

[b] We have shown that the addition of onensemble neurons or offensemble neurons does not explain the encoding precision by themselves. We have also included a groups of non-ensemble coding neurons (nonparticipant-tuned populaiton). The actual ensembles, activity patterns including both active and silenced neurons reached the highest precision. We have showed the results in Figure 5 and have added the following sentences to the results:

“Ensembles exhibit higher precision predicting drifting grating orientation than the integration of coding neurons

[...] Ensembles could have more information and a coding advantage simply because they are composed of many neurons. To investigate this, we extended our analysis to include comparisons involving the activity of different neuronal groups (Figure 3d): the onensemble (fraction of active onensemble neurons), offensemble (fraction of active offensemble neurons), and nonparticipant-tuned group (fraction of active neurons not part of the onensemble or offensemble but tuned to visual stimuli).

[...] Thus, ensembles displayed an elevated orientation selectivity and a narrow tuning width, characteristics that cannot be attributed solely to the integration of onensemble or offensemble neurons.”

5. Orientation is a circular parameter, so why did you fit the response of neurons and ensembles with a double Gaussian instead of circular wrapped gaussians or von mises? How did you solve the issue of neurons or ensembles with preferences at the edges (i.e., close to 0° or 360°)?

Response: We appreciate this observation, so we have changed the fit to a single Gaussian. We have no difficulties in detecting the tuning of a neuron at the edges since 0° and 360° grating images are the same orientation and direction.

We have modified the method to further clarify this:

“Orientation selectivity and tuning width

[...] To compute the tuning width, responses to orientations were fit by nonlinear least squares optimization to the following Gaussian curve:

$$y(\theta) = a \cdot e^{-\frac{(\theta - \theta_{pref})^2}{2 \cdot \sigma^2}}$$

where a is the maximum response amplitude of the preferred orientation, θ_{pref} is the preferred orientation, and σ is the tuning width. Coefficients were restricted to the following intervals: a to the interval [0 1], θ_{pref} to the interval [-45° 180°], and σ to the interval [1° 90°].”

6. Some results should be further discussed. [a] What is the physiological interpretation of a longer response (Fig. 2f)? [b] How do you explain the steeper calcium decays (Fig 5)? To my knowledge, the time constant of the calcium trace is only due to the physical properties of the calcium reporter...

Response: We appreciate your concern about extending our discussion.

[a] We have now included a physiological interpretation about the longer response of ensembles and added few references. We have added the following paragraph in discussion:

“Furthermore, ensembles, by combining information about both coactive and silenced neurons, may integrate information more robustly and over longer periods for perception. While individual neurons are unreliable and adapt rapidly, which could lead to shifts in perception, the reliability and prolonged activity of ensembles may help stabilize representations of visual features²⁵⁻²⁷.”

25. Carrillo-Reid, L., Han, S., Yang, W., Akrouh, A. & Yuste, R. Controlling Visually Guided Behavior by Holographic Recalling of Cortical Ensembles. *Cell* 178, 447-457.e5 (2019).

26. Marshel, J. H. et al. Cortical layer-specific critical dynamics triggering perception. *Science* (80-.). 365, (2019).

27. Dalgleish, H. W. P. et al. How many neurons are sufficient for perception of cortical activity? *Elife* 9, 1–99 (2020).

[b] We have added a brief explanation along with few more references of why calcium traces decay faster during inhibition in discussion:

“Although commonly excluded from analysis, calcium decay signals can reveal synaptic inhibition³⁶⁻³⁸, and, accordingly, we showed that calcium transients decay faster during the onset of their preferred silencing stimulus (Figure 3).

Hyperpolarization during *in vivo* Ca²⁺ imaging is known to reduce GCaMP signal levels due to the correlation between intracellular Ca²⁺ concentration and membrane polarization^{36,39,40}.”

We have added the following references:

39. Zhao, Y. Et al. Inverse-response Ca²⁺ indicators for optogenetic visualization of neuronal inhibition. *Sci. Rep.* 8, 1–8 (2018).
40. Yang, H. H. H. et al. Subcellular Imaging of Voltage and Calcium Signals Reveals Neural Processing In Vivo. *Cell* 166, 245–257 (2016).

7. I did not understand how the ROC analysis was performed. How do you use a binary classifier system to classify 8 orientations? Did you do a ROC analysis for each orientation versus the rest and take the best?

Response: In our previous approach, we used only ROC analysis, treating each individual neurons or ensemble as a binary classifier to predict its respective stimulus orientation. However, in our revised methodology, we have extended our evaluation to predict all four orientations simultaneously using multiclass confusion matrices. This is reflected in our new Figure 6 and text in the methods section, described as follows:

“Accuracy of prediction using multiclass confusion matrices

[...] For each neuron, we averaged its response during each trial and built a Receiver Operator Characteristic (ROC) curve by varying the threshold to its responses. This allowed us to evaluate its performance in predicting its tuned orientation. We selected the threshold that optimized performance on the ROC curve, maximizing the informedness (hit rate minus false alarm rate). Once we obtained the predicted orientation using the four neurons for every trial, we built the confusion matrix.”

8. In Figure 2e, how do you explain that the visually evoked response starts at t=0?

Response: We appreciate your observation. The visual response starting at t = 0 may be appear abrupt, and we acknowledge that the precise timing in calcium imaging recordings can be influenced by the filtering applied during signal preprocessing. To clarify, we have included a note in results:

“Ensembles exhibit higher precision predicting drifting grating orientation than the integration of coding neurons

[...] Note that the precise timing of activation at the stimulus onset and the duration of activity may experience slight shifts or smoothing due to the filtering applied during the Ca²⁺ signal preprocessing for spike inference, potentially leading to minor inaccuracies.”

9. In Figure 6, it would be valuable to have an example of an ensemble showing all its neurons and their tuning curves.

Response: We appreciate the suggestion for visual clarification. To address this, we have updated this figure to **Figure 7**. This revised figure now includes a raster displaying sorted subset populations for enhanced clarity. Additionally, we have added **Supplementary Figure 5**, which provides an illustrative example from one of the experiments. In this supplementary figure, we depict the responses of all onensemble, offensemble, and nonparticipant neurons, categorized into subsets based on their individual tuning properties. We believe that polar plot visualizations of individual neuronal responses offer greater informativeness compared to tuning curves, as tuning curves can sometimes suffer from suboptimal fitting (specifically for “unspecific”, “interstim”, and “untuned” neurons).

10. Line 172: Comparing a fraction of active neurons and the percentage of time the ensembles are being active seems unfair given the different nature of the measures. Why do you expect the informedness to be comparable?

Response: Thank you for raising this observation. We acknowledge the different nature of the measures. Neuronal activity and ensemble activity are binary signals, and signals from the groups of neurons (onensemble, offensemble, and nonparticipant-tuned population) are nonbinary. However, we think it is a fair comparison to compute the average of their activity regarding the signals are binary or not. And no we have provided a more detailed explanation in the methods section. Here is the added text:

“Onensemble, offensemble, and nonparticipant-tuned population activity

For each ensemble, we computed the activity of its own onensemble and offensemble, and the subset of tuned neurons within the nonparticipant population (nonparticipant-tuned population). The activity of these groups was computed by counting the fraction of their active neurons at every time point, i.e., the average of neuronal binary activity within the group.

Orientation selectivity and tuning width

The response for each neuron, ensemble, onensemble, offensemble, and the nonparticipant-tuned population was computed by averaging their activity during each visual stimulation trial. The signals of neurons and ensembles are binary, individual neuronal activity represents spiking activity (0 = inactive; 1 = active), and ensemble activity represents ensemble occurrences (0 = no occurrence; 1 = occurrence). Thus, the average of neuronal and ensemble activity represents the fraction of time they were active during the two-second visual stimulation. The signals of onensembles, offensembles, and nonparticipant-tuned populations are nonbinary with values ranging between 0 and 1, each group represents the fraction of their active neurons at any sample (0 = no neurons of the group are active; 1 = 100 % neurons of the group are active). Thus, the average of onensemble, offensemble, and nonparticipant-tuned population activity represents the average fraction of active neurons within the group during the two-second visual stimulation.”

11. Line 194: How are the categories “interstim” and “untuned” mutually exclusive?

Response: We would like to clarify that the categories “interstim” and “untuned” are indeed mutually exclusive, as stated in our methodology. We have revised the text in the methods section to clarify it. Here is the revised text:

“Single-cell tuning to multiple orientations, interstimulus periods, or untuned

[...] samples were obtained measuring the average neuronal activity during two different periods: stimulation periods (independently of the orientation) and interstimulus periods. These samples were evaluated using a two-sample t-test, where significant neurons were labeled as “unspecific” if they were more active during stimulation periods and were labeled as “interstim” if they were more active during interstimulus periods. Nonsignificant neurons were label as “untuned” neurons. Using this test, we identified significantly active neurons during visual stimulation (unspecific), significantly active neurons during interstimulus periods (interstim), and neurons that did not exhibit significant changes in their activity between visual stimulation and interstimulus periods (untuned).”

12. As a gaussians is fit on the responses of the neurons to the presentation of stimuli of 8 possible orientations, the preferred orientations can take any value between 0°-360°. What are the criteria for the decision that a neuron is tuned to the preferred orientation of an ensemble? This is particularly important to understand how you classify neurons within ensembles and offsembles.

Response: We used orientation selectivity to assign the preferred orientation, which falls under one of the four orientations. We have clarified this in the methods:

“Orientation selectivity and tuning width

[...] Orientation circular variance ($1 - \text{CirVar}$) was used to quantify orientation selectivity and the Hotelling’s t^2 -test was used to assess significance²³. Significant orientation selectivity was used to assign tuning to neurons and ensembles, with each being assigned to one of the four presented stimulus orientations. Once identified tuned ensembles, we proceeded to extract the activity of their own onsemble, offsemble, and nonparticipant population and measure the orientation selectivity for each group.”

Minor comments

1. Line 147: Please rephrase this sentence, the reader cannot understand what is compared to what.

Response: Thank you for asking clarification. We have revised the entire results section. Specifically, we mention that offsembles contain more neurons than onsembles, and that only a fraction of onsemble neurons were tuned to the same orientation of the ensemble. Here is the text added to results:

“Patterns of neuronal activation and silencing during stimulus presentation

[...] Offensebles comprised a larger proportion of neurons compared to onensebles, with an average of 139 ± 11 offenseble neurons versus 108 ± 9 onenseble neurons for each ensemble ($20 \pm 1\%$ and $26 \pm 2\%$ of the total neurons, respectively).

[...]

Ensembles are composed of neurons with diverse tuning properties

[...] In terms of onenseble composition, on average there were 44 ± 3 neurons tuned to the same stimulus orientation [...] ($41 \pm 3\%$ [...] of onenseble neurons [...]). Thus, onensebles consist not only of coactive neurons tuned to the same orientation, but also of coactive neurons tuned to different or multiple orientations.”

2. Line 422 : zero, not cero

Response: We have corrected this, thank you!

Thank you again, these revisions have significantly improved the manuscript's clarity and validity, further strengthening the novel findings we present.

Reviewer #2 (Remarks to the Author):

Pérez-Ortega et al. examine orientation tuning ensembles (to first approximation, neurons with highly correlated activity) in L2/3 of V1 in GCaMP-expressing transgenic mice. Ensembles are detected using a custom algorithm. They then define an "ensemble participation index" which can detect not just ensemble neurons but also neurons that are effectively anti-correlated with the ensemble, or "offensembles". They show that decoding of orientation is better done when you combine information from both the ensemble and the offensembles. The idea is interesting and well presented and the experimental methodology is of high quality. The statistical analysis, however, is done improperly, basically because they pool across thousands of neurons in most cases which trivially makes P tiny. Given that n=7 mice (and 5 in some cases), it's likely but not certain that the existing data will suffice to do the statistics correctly. Given this situation, I do not feel confident in evaluating the other claims of the paper, since I simply do not know to what extent they are / are not valid, and will only comment on clarity/methodological issues in my review.

Thank you for your comprehensive review, and we highly value your feedback. We have taken your comments into careful consideration and made significant improvements to our methodology, including conducting new experiments to enhance the robustness of our findings by adding statistical power.

Major:

1. My largest concern is the statistical approach. To my understanding, in many analyses, the authors aggregate neurons across animals. Hence the $P < 10^{-78}$ values we see in some of the figures. Aggregating neurons across animals and treating them as independent observations is improper because within animal, neurons co-vary, and strongly. Indeed, correlated and anti-correlated neurons are at the heart of this paper's claims, making aggregation particularly problematic. Aggregating is also bad because it is strongly biased by, potentially, a single animal with very large numbers of neurons or with a very large effect. I recommend the following papers for the authors, as they outline the problem nicely and succinctly: Aarts et al., 2015 Nat. Neurosci.; Aarts et al., 2014 BMC Neurosci; Galbraith et al. 2010 J Neurosci.

As explained in all the referenced papers, the appropriate approach is to either use animal-wide summary statistics (e.g., for Fig. 2C, compute the mean 1-CirVar for each group - neurons or ensembles - in each animal, then you will have n=7 measurements for tuned neurons and tuned ensembles) or a multi-level statistical analysis (this will slightly improve things beyond an n=7 - you may end up with power that you would get from 10 samples, for instance - but not thousands).

This problem is unfortunately present in almost every analysis - by figure panel, 2c, 2e, 2f, 2h, 2j, 3f, 4f, 5c, 6,d, etc. .

The one place this was a problem was not present was Fig 4e, where per-animal statistics were correctly used. Unfortunately, a Mann Whitney test was used with n=5 and no effect was found, from which the conclusion was drawn that locomotion had no effect. You will never have $p < 0.05$ with n=5 for Mann-Whitney. The test is inconclusive and the "conclusion" that things were not different between still and running cannot be made from the data.

Response: We thank you for providing us with feedback that has allowed us to improve our statistical analysis and confirm the robustness of our results. We have performed new experiments to increase the statistical power and we now have 12 experiments in total. We have used animal-wide statistics, so each animal is a single averaged measure in all of our comparisons.

We have added the following text in methods:

“Animal-wide summary statistics

All statistical tests were performed using animal-wide summary statistics to assure independent observations and avoid nesting data⁵⁹⁻⁶¹. Every single data point in boxplots represents the average within a mouse and is paired in different comparisons (Figure 2e, Figure 3c, Figure 4e-f, Figure 5c,e,g, and Figure 6b). Traces and shaded regions represent the mean and SEM across mice (Figure 2g and Figure 5b,f). We averaged single neuron measures per stimulus orientation tuned, and then averaged again to get a single datapoint per mouse. For ensembles, we averaged the values of all tuned ensembles to get a single datapoint per mouse. Similarly, for onsembles, offsembles, and nonparticipant populations, we averaged the values to get a single datapoint per mouse.”

We have cited the recommended papers:

59. Aarts, E., Verhage, M., Veenvliet, J. V., Dolan, C. V. & Van Der Sluis, S. A solution to dependency: Using multilevel analysis to accommodate nested data. *Nat. Neurosci.* **17**, 491–496 (2014).
60. Aarts, E., Dolan, C. V., Verhage, M. & Van der Sluis, S. Multilevel analysis quantifies variation in the experimental effect while optimizing power and preventing false positives. *BMC Neurosci.* **16**, 1–15 (2015).
61. Galbraith, S., Daniel, J. A. & Vissel, B. A study of clustered data and approaches to its analysis. *J. Neurosci.* **30**, 10601–10608 (2010).

2. This brings me to problem 2: in most of the statistical tests, n is not given, and it is not explained that all neurons/ensembles were aggregated across animals. Specify the n for every statistical test and the unit of n - and in the resubmission, this should in almost all cases be mice. If it is not mice, an appropriate multielevel test should be used and explained in detail in the methods.

Response: We have added the number of animals used on each statistical test. We now used a single datapoint per mouse in all cases.

3. Behavior is undercharacterized/underdescribed. It would be very useful to know what the animal is doing during most of the experiment. We get a hint of this in Fig.4 when locomotion is examined but as a reader I want to know the general statistics of behavior - how frequently is the mouse moving, are there long periods of standing still, does the animal groom, etc. Are animals being monitored for wakefulness? Or is running sufficiently

frequent that the animal is clearly not asleep (if so, describe / show this). How frequently, in general, do animals move and run? Is the experimenter monitoring the animal with video?

Response: We truly appreciate this input as it compelled us to further characterize mouse behavior in our new experiments. We now include mouse face recordings in our analysis to characterize mouse behavior. We performed experiments until mice became fatigued. We extended our experiments up to one hour post visual stimulation to include a large enough subset of distinct facial movements (i.e., changes in the eye opening). We measured additional behaviors (running speed, whisking, sniffing, and blinking) and added a new supplementary figure (Supplementary Figure 1).

We added the following text to the results:

“Patterns of neuronal activation and silencing during stimulus presentation

[...] Mouse running speed and facial behaviors, recorded with an infrared camera, were tracked to measure wakefulness. Whisking behavior, reflecting mouse wakefulness or engagement, accounted for 61 ± 3 % of the experiment duration. In contrast, when mice exhibited signs of disengagement or fatigue, they only whisked for 34 ± 4 % of the time (Supplementary Figure 1).”

We also added a new method:

“Mouse facial recording

Mouse face was recorded using an infrared monochrome camera (DMK 21BU04.H, The Imaging Source) with a zoom lens (MVL7000, Navitar) and an infrared illuminator (AI4, Tendelux). Images were acquired at 30 frames per second and stored using IC Capture software (The Imaging Source). Whisking, blinking, and sniffing behaviors were measured using a custom-made MATLAB code⁵⁷.”

We have cited our source code:

57. Pérez-Ortega, J. MoussionEnergy. (2023). doi:10.5281/zenodo.8422691

Minor:

- Missing information: what is the threshold for ensemble membership. In the methods all that is said is that significant must be met with z-test on permuted data. Is this with a $p < 0.05$ threshold?

Response: Thank you for pointing this out. We have added that the threshold that we used to detect membership is $p < 0.05$ in the methods.

We have revised our method as follows:

“Identification of onsemble and offsemble neurons

We introduced the ensemble participation index (EPI) to measure each neuron's involvement during ensemble activity, ranging from -1 to 1 . A positive EPI indicates increased activity during ensemble activity, while a negative EPI indicates reduced activity. We determined EPI significance ($p < 0.05$) using a two-sample t-test, comparing the fraction of time that the neuron was active during ensemble activity vs ensemble inactivity. Neurons with significant positive EPIs were classified as “onsemble” neurons, and those with significant negative EPIs as “offsemble” neurons.”

Once again, we want to express our sincere appreciation for your invaluable feedback to help us improve our research. These revisions have significantly enhanced the quality and rigor of our work.

REVIEWER COMMENTS

Reviewer #1 (Remarks to the Author):

I would like to thank the authors for their effort in improving their manuscript. Despite the manuscript extensive revisions, I still have outstanding issues with some of the main claims of the report.

1. The title does not accurately reflect the manuscript content. The authors are not demonstrating that neurons of the offsemble (the silenced neurons) are actively silenced. This remains a hypothesis mentioned in the discussion (line 328). Moreover, the demonstration that the presence of offsemble increases the encoding precision remains to be provided (as the authors write line 209: “our finding suggests”). Indeed, Fig 6b does not demonstrate that the inclusion of offsemble neurons is responsible for the difference in precision between ensembles and onsembles.

2. The results presented Fig.5 still present significant issues. The authors compare binarized data (neurons and ensemble) with continuous data (fraction of active neurons). The binarization (active/ not active) artificially increases the signal-to-noise ratio by thresholding the noise. The tuning width and orientation selectivity indexes presented in Fig. 5e,g measure a signal to noise ratio (preferred vs. non-preferred). For a fair comparison, I would use the % of ensemble neurons participating in the response (either by being active or being actively not active for the onsemble and offsemble neurons respectively). Currently, the demonstration of the better orientation selectivity of ensemble compared to onsemble or offsemble is not made.

3. Line 137: What is the preferred orientation of an offsemble? Is it the orientation for which offsemble neurons are the most suppressed? If it is the case, is the term “preferred orientation” adapted?

4. Line 245-246. I do not understand what you mean by: “distributed processing” and “coordinated activity of ensemble and offsemble”. Where can I see that in the data?

5. Line 249: What do you mean by “due to its heterogeneous neuronal composition”?

6. The title of Fig. 6 is not accurate. Neurons in V1 would “predict” the orientation of the stimulus if they would encode the stimulus before this one occurs.

Reviewer #2 (Remarks to the Author):

I thank the authors for the thorough job in the resubmission. The paper is much improved and, in my view, is ready for publication. Congratulations to the authors for an interesting study.

Response to Reviewers

Nature Communications manuscript NCOMMS-23-12936A

We are grateful for the time and effort dedicated to reviewing our manuscript. In consideration of the feedback received, we have made the following revisions:

Title and abstract: We have changed the title to remove the claim that neurons are actively silenced and we have revised the abstract accordingly.

Results and Methods Sections: We have revised the text in results and methods to enhance the clarity, added new results, and added a new method.

Discussion: We have scaled down our claims of inhibition.

Figures: We have revised the legend title on Figure 6, extended our Figure 5, and added three new Supplementary Figures.

REVIEWER COMMENTS

Reviewer #1 (Remarks to the Author):

Reviewer #1 (Remarks to the Author):

I would like to thank the authors for their effort in improving their manuscript. Despite the manuscript extensive revisions, I still have outstanding issues with some of the main claims of the report.

Response: Thank you very much!

1. The title does not accurately reflect the manuscript content. The authors are not demonstrating that neurons of the offsemble (the silenced neurons) are actively silenced.

Response: We have changed the title removing the claim that the neurons are “actively” silenced. The new title is “Neuronal offsembles: stimulus encoding by inactivation of cortical neurons” and we refer to their lack of firing as “inactivation”, a term that is agnostic as to the mechanism.

This remains a hypothesis mentioned in the discussion (line 328).

Response: We have rewritten the abstract clarifying that the inhibitory silencing is a hypothesis that needs direct confirmation. Accordingly, in the manuscript we also have replaced the words “actively silenced”, “inhibition”, and “inhibited” by “inactive”, “inactivation”, and “inactivated”.

The new abstract reads:

“Neuronal ensembles are groups of coactive neurons associated with motor, sensory, and behavioral functions. However, how ensembles encode information remains unclear. To explore this, we investigated the responses of layer 2/3 visual cortical neurons in awake mice using two-photon volumetric calcium imaging during visual stimulation. We identified neuronal ensembles employing an unsupervised model-free algorithm. In response to visual stimuli, ensembles exhibited small trial-to-trial variability and high orientation selectivity. During ensemble occurrence, besides neurons that were significantly activated by the stimulus, we also found neurons whose activity was significantly decreased. To distinguish between these two groups of neurons, we introduce the terms “onsemble” for the significantly active neurons and “offsemble” for the inactive ones. Ensembles demonstrated superior predictive accuracy for visual stimulus orientation compared to summing the activity of individual onsemble or offsemble neurons. Interestingly, calcium decay kinetics in offsemble neurons became faster during stimulus presentation, as if offsemble neurons were selectively inhibited. We conclude that the combined activation and inactivation of onsemble and offsemble neurons enhances visual encoding and hypothesize that the inactivation is due to inhibitory interneurons. Ensembles could represent functional units of information in cortical circuits, combining selectively neuronal activation and inactivation as an emergent and distributed neural code.”

Moreover, the demonstration that the presence of offsemble increases the encoding precision remains to be provided (as the authors write line 209: “our finding suggests”). Indeed, Fig 6b does not demonstrate that the inclusion of offsemble neurons is responsible for the difference in precision between ensembles and onsembles.

Response: We have addressed the concern about the necessity of offsemble neurons for encoding precision with a different analysis. We now demonstrate that, if we remove offsemble neurons from the analysis of ensemble identification, the ensemble orientation selectivity is significantly reduced. This does not happen when we remove the same number of nonparticipant neurons. We have added a new Supplementary Figure 7.

We have added the following paragraph to the results:

“Offsembles enhance orientation encoding of ensembles

Finally, to test if offsemble neurons are responsible for the difference in precision between ensembles and onsembles, we reanalyzed the population activity removing offsemble neurons. Using the same algorithm to identify neuronal activity patterns, we found fewer ensembles encoding orientations, a significant reduction in their orientation selectivity and a broader tuning width (Supplementary Figure 7). This effect did not happen when we removed the same number of nonparticipant neurons, even when the average of neurons removed was $60 \pm 3\%$ (mean \pm SEM across 12 mice). Thus, the inclusion of offsembles neurons enhanced the precision

of the ensemble. In summary, ensembles are highly accurate in predicting drifting grating orientations. While it might have been tempting to attribute this precision solely to the active neurons during ensemble activation (ensembles), our findings suggest that ensemble high precision was essentially due to the emergent participation of both onsembles and offsembles.”

2. The results presented Fig.5 still present significant issues. The authors compare binarized data (neurons and ensemble) with continuous data (fraction of active neurons). The binarization (active/ not active) artificially increases the signal-to-noise ratio by thresholding the noise.

Response: We have reanalyzed neuronal activity to identify neuronal ensembles using the spike inference for each neuron (nonbinary neuronal data) and measured the orientation selectivity and tuning width. This analysis confirms our results. We have included this results in a new Supplementary Figure 4.

We added the following sentence in Results:

“Ensembles have higher precision predicting orientation than individual neurons

[...] To test if the binary spike detection biased the higher selectivity of ensembles, we repeated the analysis by measuring orientation selectivity and tuning width using spike inference data and obtained similar results (Supplementary Figure 4).”

The tuning width and orientation selectivity indexes presented in Fig. 5e,g measure a signal to noise ratio (preferred vs. non-preferred). For a fair comparison, I would use the % of ensemble neurons participating in the response (either by being active or being actively not active for the onsemble and offsemble neurons respectively). Currently, the demonstration of the better orientation selectivity of ensemble compared to onsemble or offsemble is not made.

Response: We have addressed this concern by using the measure suggested by the reviewer (fraction of ensemble neurons participating in the response). The new analysis confirms our results. We have extended our Figure 5 and added a new Supplementary Figure 5 to show the ensemble response by counting the proportion of activated neurons from onsemble and the inactivated neurons from the offsemble.

We have added this paragraph in the Results:

“Ensembles have higher precision predicting orientation than individual neurons

[...] To investigate this, we measured the proportion of active neurons in ensembles (fraction of active onsemble neurons), offsembles (fraction of active offsemble neurons), ensembles (fraction of activated onsemble and inactivated offsemble

neurons from onensemble and offensemble, respectively) and nonparticipant-tuned neurons (fraction of active neurons not part of the ensemble but tuned to visual stimuli; Figure 5d). We compared the orientation selectivity and tuning curves between all of these groups. Ensembles, both as binary categories or as the fraction of participating ensemble neurons, exhibited a high orientation selectivity of 0.98 ± 0.01 and 0.61 ± 0.05 , respectively, than individual neurons (0.46 ± 0.01). Interestingly, the orientation selectivity of onensembles (0.23 ± 0.02) and offensembles (0.20 ± 0.01) was lower than that of individual neurons. [...] Similarly, when fitting a Gaussian tuning curve to each group, ensembles showed a narrower bandwidth (binary: $8^\circ \pm 1^\circ$; fraction of participating ensemble neurons: $29^\circ \pm 3^\circ$) than individual neurons ($37^\circ \pm 1^\circ$). Onensembles, offensembles, and nonparticipant-tuned group displayed broader tuning ($52^\circ \pm 2^\circ$, $55^\circ \pm 1^\circ$, and $80^\circ \pm 2^\circ$, respectively; means \pm SEMs across 12 mice; Figure 5f-g). [...] Additionally, during the preferred stimulus orientation, the proportion of responding ensemble neurons was higher ($78 \pm 2\%$) than the proportion of tuned neurons ($49 \pm 2\%$) and the proportion of responding onensemble neurons ($47 \pm 2\%$; means \pm SEMs across 12 mice; Supplementary Figure 5). Therefore, the simultaneous activation and inactivation of neurons enhance the ensemble orientation selectivity and a narrow tuning width.”

3. Line 137: What is the preferred orientation of an offensemble? Is it the orientation for which offensemble neurons are the most suppressed? If it is the case, is the term “preferred orientation” adapted?

Response: The preferred orientation of an offensemble is the same as that of the ensemble to which it belongs. In other words, the preferred orientation is computed for each ensemble, which is where the ensemble has more occurrences..

We have clarified this in the Results:

“Offensemble neurons are inactivated

[...] The preferred orientation of an offensemble is the same as that of the ensemble to which it belongs.”

4. Line 245-246. I do not understand what you mean by: “distributed processing” and “coordinated activity of ensemble and offensemble”. Where can I see that in the data?

Response: Distributed processing refers to the encoding across multiple neurons, which collaboratively perform specific tasks, such as representing visual stimulation. An ensemble is a distributed pattern of activated neurons (onensemble) and inactivated neurons (offensemble), and this pattern represents accurately a visual stimulus orientation. This is clearly visualized in Figure 1d, Figure 2a,f, and maybe a better visualization is in Figure 7a.

5. Line 249: What do you mean by “due to its heterogeneous neuronal composition”?

Response: Heterogeneous composition refers to a composition of multiple neurons with different individual functional roles that emerge in a coordinated activity only during an specific visual stimulation and no other. We have edited the Results:

“Ensembles neurons have diverse tuning properties

[...] In conclusion, our study reveals that cortical circuits respond to visual stimuli through a simultaneous activation and inactivation of multiple neurons with diverse individual tuning properties. Thus, ensemble coding represents an emergent property, activating onsemble neurons and inactivating offsemble neurons, that provides reliable and highly precise representation of visual information.”

6. The title of Fig. 6 is not accurate. Neurons in V1 would “predict” the orientation of the stimulus if they would encode the stimulus before this one occurs.

Response: We have changed the title of Figure 6 to “Ensembles encode orientation better than onsembles or offsembles”.

Reviewer #2 (Remarks to the Author):

I thank the authors for the thorough job in the resubmission. The paper is much improved and, in my view, is ready for publication. Congratulations to the authors for an interesting study.

Response: Thank you very much, we really appreciate it!

REVIEWERS' COMMENTS

Reviewer #1 (Remarks to the Author):

I do not have further comments

Reviewer #1 (Remarks on code availability):

I have tested the code on my own data. I obtained satisfactory results.